# Do AI Models Perform Human-like Abstract Reasoning Across Modalities?

## Abstract

OpenAI's o3-preview reasoning model exceeded human accuracy on the ARC-AGI benchmark, but does that mean state-of-the-art models recognize and reason with the abstractions that the task creators intended? In this work, we investigate the abstraction abilities of AI models using the ConceptARC benchmark. We evaluate models under settings that vary the input modality (textual vs. visual), whether the model is permitted to use external Python tools, and, for reasoning models, the amount of reasoning effort. In addition to measuring output accuracy, we perform fine-grained evaluation of the natural-language rules that models generate to explain their solutions. This dual evaluation allows us to assess whether models solve tasks using the abstractions that ConceptARC was designed to elicit, rather than relying on surface-level patterns. Our results show that, while some models using text-based task representations match human output accuracy, the best models' rules are frequently based on surface-level "shortcuts", and capture intended abstractions substantially less often than do humans. Thus their capabilities for general abstract reasoning may be overestimated by evaluations based on accuracy alone. In the visual modality, AI models' output accuracy drops sharply, yet our rule-level analysis reveals that models might be underestimated, as they still exhibit a substantial share of rules that capture intended abstractions, but are often unable to correctly apply these rules. In short, our results show that models still lag humans in abstract reasoning, and that using accuracy alone to evaluate abstract reasoning on ARC-like tasks may overestimate abstract-reasoning capabilities in textual modalities and underestimate it in visual modalities. We believe that our evaluation framework offers a more faithful picture of multimodal models' abstract reasoning abilities and a more principled way to track progress toward human-like, abstraction-centered intelligence.

## 1 Introduction

The ability to quickly form abstractions and reason with them via analogy is central to humans' remarkable capacity to generalize knowledge to novel situations (Carey, 2011; Hofstadter, 2001; Lake et al., 2017). Many benchmarks have been designed to evaluate abstract reasoning abilities in machines (Foundalis, 2025; Hofstadter, 1995; Zhang et al., 2019). Among the most prominent such benchmarks is the Abstraction and Reasoning Corpus (ARC) (Chollet, 2019). ARC consists of a set of idealized problems that require few-shot rule-induction and analogical reasoning. As Figure 1 shows, each puzzle ("task") consists of a small set of *demonstrations*—initial and transformed grids—and a *test* grid, each ranging in size from $1 \times 1$ to $30 \times 30$, with each cell having one of 10 possible colors. To solve a task, an agent should infer a rule governing the demonstrations and apply that rule to the test input to produce a correct output grid.

Chollet 2025 devised 1,000 such tasks, releasing 400 easier puzzles as a "training set," 400 harder puzzles as an "evaluation set," and keeping the remaining harder puzzles to form private test sets. Participants in the 2024 ARC-AGI Prize competition entered programs to vie for monetary prizes, including a $600,000 grand prize for a program that exceeds 85% accuracy—that is, percentage of correct output grids—on a private test set of 100 tasks[1]. The top scoring program, which employed a fine-tuned LLM and extensive data augmentation, reached about 54% accuracy (Chollet et al., 2024).

---

[1] Average human performance was measured on the comparable, but somewhat easier public evaluation set as 64% (LeGris et al., 2024).

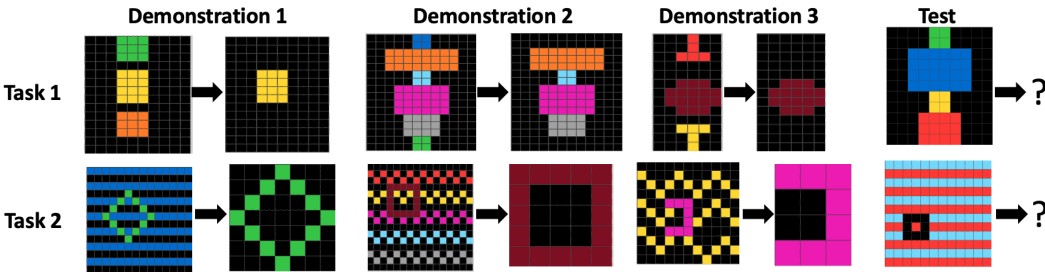

Figure 1: Each row shows a task from the ConceptARC benchmark. Each task shown consists of three demonstrations of a transformation and one test grid. In this study, the solver is tasked with generating a rule that describes the transformations and applying that rule to the test grid.

After the competition, Chollet and colleagues, with collaboration from OpenAI, tested a pre-release version of OpenAI's o3 model on a different "semi-private" test set of 100 tasks. This model achieved 76% accuracy on its low-effort setting and 88% accuracy on its high-effort setting, with computing cost per task estimated at $200 and $20,000 respectively (Chollet et al., 2025). While o3-preview was not qualified to participate in the official competition,[2] its superior performance was described as "a genuine breakthrough, marking a qualitative shift in AI capabilities compared to the prior limitations of LLMs"(Chollet, 2024). However, despite the high accuracy of o3 on ARC tasks, it is not clear to what extent AI systems have achieved human-like abstract reasoning abilities. Consider the task illustrated in the top row of Figure 1. A human solving this task is likely to be able to generalize across different instantiations of the underlying abstract concepts—identifying and removing the top and bottom objects—no matter the size, shape, color, position, or number of objects. To our knowledge, no prior studies have assessed whether AI systems such as o3 are solving these tasks by using the intended, generalizable abstractions, or if they are inferring less generalizable rules ("shortcuts") based on unintended correlations in task demonstrations.

Here we assess the abstractions used by several commercial and open-weight models in solving tasks from ConceptARC (Moskvichev et al., 2023), a benchmark in the ARC domain containing tasks organized around basic spatial and semantic concepts, such as "inside vs. outside," "above vs. below," "extend to boundary," and "same vs. different." For example, the tasks shown in Figure 1 are from ConceptARC's "top vs. bottom" and "extract object" concept groups, respectively. As described in Moskvichev et al. (2023), ConceptARC was designed to test robust understanding of these concepts by providing tasks—designed to be simple for humans—that deploy each concept in varying contexts and require varying degrees of generalization. Because it isolates simple abstract concepts, we believe this benchmark to be better suited than the original ARC dataset for investigating the concepts used by humans or machines in solving tasks. Although there is likely a big overlap in concepts used in both datasets, ARC frequently employs compositional reasoning, which would make it necessary to disentangle different types of intended abstractions, complicating the evaluation process. A similar evaluation on the original ARC dataset is an interesting avenue for follow-up work.

Previous evaluations using the o3 model (as well as all entries in the 2024 ARC-AGI Prize competition) relied on text-based representations of the demonstration and test grids to solve each ARC task. Each grid is represented as an integer matrix, with entries encoding colors indexed from 0 to 9. However, o3 and related models are reported to possess sophisticated reasoning abilities in both textual and visual modalities (OpenAI, 2025). In our experiments, we investigate the models' abstract reasoning abilities in both modalities. We also examine how reasoning effort (the token budget allocated for the reasoning stage) and access to external "tools" (here, the ability to generate and execute Python code) affect a model's ability to discover abstract rules and solve tasks.

In the following sections, we describe our experimental setup and results, and discuss how our findings relate to three central questions: (1) How does the accuracy achieved by AI models on ConceptARC tasks compare to that of humans? (2) To what extent do the rules generated by AI

---

[2]OpenAI's o3 model violated two key competition rules: it could not be run locally on the competition server and its processing required substantially more than the required time limit.

models and by humans capture the abstractions intended by the test designers, and to what extent do they rely on unintended, superficial patterns? (3) How do modality (textual vs. visual), reasoning effort (token budget), and Python tool access affect how well models can solve these tasks via the intended abstractions?

## 2 METHODOLOGY

**Dataset and Experiments**  To create ConceptARC, Moskvichev et al. 2023 chose 16 basic spatial and semantic concepts, and for each concept created 30 tasks that focused on that concept in different instantiations, with different degrees of abstraction, for a total of 480 tasks. Unlike the original ARC corpus, all ConceptARC tasks were designed to be relatively easy for humans, since each relies on a simple abstract concept and a straightforward application of that concept to a novel test grid.

We evaluated four proprietary multimodal "reasoning" models on the ConceptARC tasks: OpenAI's o3 and o4-mini, Google's Gemini 2.5 Pro, and Anthropic's Claude Sonnet 4. For comparison, we also evaluated three non-reasoning multimodal models: OpenAI's GPT-4o, Meta's Llama 4 Scout, and Alibaba's Qwen 2.5 VL 72B. To maximize reproducibility, non-reasoning models were run with temperature 0. Because the o3, o4-mini, and Claude Sonnet 4 APIs restrict temperature to 1.0, we used temperature 1.0 for all four reasoning models to maintain comparability. For experiments using the textual modality, we used the same prompt as in Chollet et al.'s 2024 evaluation of o3-preview. For experiments using the visual modality, we used a slightly modified version of this prompt. These prompts are given in full in Appendix A and Appendix B.

For both modalities, models were asked to generate a JSON object containing the transformation rule and the corresponding output grid, represented as a matrix of integers. This setup enables two-fold evaluation: (i) grid output accuracy and (ii) the degree to which model-generated rules capture the tasks' intended abstractions. We evaluated human-generated solutions to these tasks using the same criteria, analyzing unpublished data[3] obtained from the study reported by Moskvichev et al. 2023 in which humans (participants on the Prolific Academic platform) were presented with ConceptARC tasks as images and asked to produce both the correct output grids and the rules they used to generate them. For each model setting, each task was given in an independent prompt, with the context window reset (cleared) before a new task was given. Due to resource constraints, we report pass@1 results for both AI models and humans.[4]

We evaluated o3 under its low- and medium-effort reasoning settings.[5] We evaluated Gemini 2.5 Pro and Claude Sonnet 4 with a reasoning budget of 16,000 tokens, which roughly approximates OpenAI's medium-effort setting. Additionally, for reasoning models we evaluated two tool-access conditions: one in which Python tools were enabled and one in which they were not.

**Evaluating Responses of Humans and AI Models**  Evaluating output-grid accuracy in human and model responses is straightforward, since each task's ground-truth solution is given in the ConceptARC corpus. For each task, humans were given the demonstrations and test grid images and were asked to generate the output grid using a custom editing tool, and models were asked to generate the output grid as a matrix with colors encoded as integers. The resulting grids can be compared automatically with the ground truth; only exact matches are considered correct.

While output-grid accuracy has been widely used to assess AI model performance on ARC tasks, to our knowledge, no prior studies have investigated whether output-grid correctness on ARC tasks reflects a grasp of the intended abstract concepts underlying the tasks, or alternatively, the extent to which correctness can be achieved by identifying and exploiting unintended, superficial patterns ("shortcuts"). It is well known that large neural-network models are capable of discovering "spurious" patterns in data and using these patterns to arrive at correct answers (Du et al., 2023; Geirhos et al., 2020). To investigate the extent to which AI models are using human-like abstractions to solve tasks, in our experiments we asked the models to output not only the transformed test grid but also a

---

[3]A. Moskvichev, personal correspondence

[4]The ARC Prize competition reported pass@2 results, that is, two independent runs on each task. If one of the runs produces a correct output grid, the task is counted as correctly solved. Moskvichev et al. 2023 reported pass@3 results for both humans and the GPT models they tested.

[5]OpenAI does not specify the token budget allocated to these settings. Due to resource constraints, we did not test the high-effort setting.

natural-language rule describing the transformation. Moskvichev et al. 2023 similarly collected such rules from their human participants, though only for correctly solved tasks.

Evaluating correctness of natural-language rules still requires human judgment.Accordingly, we manually annotated both model- and human-generated rules on their quality, distinguishing between "incorrect" (rules that do not work on the demonstrations, as well as rare cases where the model did not return a rule), "correct-unintended" (rules that work on the demonstrations and potentially on test grids, but do not capture the intended abstractions), and "correct-intended" (rules that align with the intended abstractions). For example, in the first task shown in Figure 1, a sample human-generated rule is "You will see a number of shapes in a vertical orientation. Copy the grid from the example then remove the topmost and bottom-most shapes. The middle shapes remain unchanged." We rated this rule as correct-intended. o3's generated rule, from our experiment with textual inputs and medium reasoning effort, was "remove any object that intersects either the topmost or bottommost non-empty row, replacing it with zeros," which we also rated as correct-intended. o3's generated rule in our experiment with visual input and medium reasoning effort was "Delete the highest and lowest coloured components in the grid (the first and last contiguous non-black groups when scanning from top to bottom); keep the rest unchanged." Also correct-intended, even though the output grid was incorrect in this case.

In the second task in Figure 1, a sample human-generated rule is "Take the shape that is imposed onto the background pattern and simply use that shape as the solution." We rated this as correct-intended. o3 with textual input and medium effort generated the rule "Find the single colour absent from both the first row and the first column. Crop the minimal rectangle that contains all occurrences of this colour, keep that colour and set everything else inside the crop to 0." We rated this as correct-unintended, since it correctly described the three demonstrations, but ignored the intended abstraction of extracting an object. In fact, we noticed that, given the textual representation of a task, o3 most often phrased its rules in terms of colors or individual pixels rather than objects. o3 with visual input and medium effort generated the rule "Crop the minimal bounding box around the unique 1-cell-thick closed loop. Leave the loop's colour unchanged and recolour every other cell in the crop with the most frequent colour of the original grid," which we rated as "incorrect". It is important to clarify that we do not inherently consider "correct-unintended" rules to be incorrect; "correct-unintended" simply refers to mismatch between the generated rule and the one intended by the task designer, which is based on humanlike "core knowledge" concepts Chollet (2019). We focus on distinguishing such patterns, in order to give a more meaningful estimation of human-like abstraction.

While it is not certain that the natural-language rule a model generates for a given task is a faithful description of how it solved that task, we manually analyzed the alignment between the generated rule and output grid for tasks in several experimental settings, and found that in over 90% of the cases the output grid (correct or not) was faithful to the model's rule, providing evidence that that these rules are good proxies for the reasoning process of the model. More details are given in the next section.

## 3 RESULTS

**Output Grid Accuracy**   Table 1 and Table 5 (in the Appendix) give the pass@1 output-grid accuracies of the reasoning models and non-reasoning models we evaluated in both textual and visual modalities.[6] In all cases, non-reasoning models attain much lower accuracy than reasoning models, so here we focus our analysis on the reasoning models. For all models, we see a dramatic performance gap between the textual and visual settings. Further, especially for o3 and o4-mini, and to a lesser extent for Claude and Gemini, we observe a jump in visual accuracy when Python tools are enabled. In contrast, allowing Python tools does not have a similar effect in the textual setting for three of the models, with o4-mini being the only exception. For o3 and o4-mini, increased reasoning effort is associated with increased accuracy in the textual modality, with or without tools; in the visual modality, we observe that the models primarily use the increased reasoning budget to execute more Python code, which may explain the substantial improvement in medium effort + tools.

---

[6]Following the ARC-Prize evaluation ARC-Prize (2024), we counted an output grid as correct only if it perfectly matched the ground truth and the requested format. For a more detailed analysis of format deviations see Appendix I.

Table 1: **Reasoning models:** Output-grid accuracy (pass@1) for Concept-ARC across models and experimental settings. Accuracy is shown in %. Each cell shows *textual / visual* accuracy. For o3 and o4-mini, we use the "low" and "medium" effort settings in the OpenAI API. For Claude and Gemini, we use a 16K reasoning token budget to approximate o3's medium effort setting. Temperature is set to 1 for all models. Bold numbers correspond to the highest visual and textual scores in each column.

| Reasoning model | low effort | medium effort | low effort + tools | medium effort + tools |
|---|---|---|---|---|
| | Textual / Visual | Textual / Visual | Textual / Visual | Textual / Visual |
| **o3** | **68.3** / 6.7 | **77.1** / 5.6 | **67.9** / 18.1 | 75.6 / **29.2** |
| **o4-mini** | 52.1 / 3.8 | 70.8 / **8.1** | 57.3 / 6.7 | **77.7** / 25.0 |
| **Claude Sonnet 4** | N/A | 60.2 / 5.2 | N/A | 55.0 / 6.9 |
| **Gemini 2.5 Pro** | N/A | 66.0 / 4.2 | N/A | 60.4 / 5.8 |

Inspecting the failure cases of the visual setting more closely, we find that models struggle to recognize the correct grid size from the image inputs. When Python tools are enabled, the models use computer vision libraries to partially compensate for this difficulty. In both textual and visual modalities, the majority of incorrect output grids are due to a simple mismatch between the generated and ground-truth grids, but—particularly in the visual setting—there is a small share of invalid outputs, either due to uneven row lengths or non-integer tokens in the grid. Figure 6 gives an error-type distribution for o3.

Using unpublished data from Moskvichev et al.'s 2023 study, we found that human-generated output grids achieved an overall pass@1 accuracy of **73%** on the 480 ConceptARC tasks, lower than that of the top reasoning models in the textual modality. (We provide per-concept accuracies in Appendix F.)

**Rule Evaluation**    Our team manually evaluated the rules generated by o3 in all settings and by Claude Sonnet 4 and Gemini 2.5 Pro in the medium-effort + tools setting, for both textual and visual modalities. We also evaluated the pass@1 rules generated by humans, using data from the study by Moskvichev et al. (2023). For each rule (human or machine-generated), an initial classification was assigned by one member of our team, and reviewed by a second member. For each rule for which there was disagreement or uncertainty about its classification, our team discussed the rule together until we came to a consensus. Due to low accuracy of other settings and resource limitations of our team, we did not evaluate rules from other models or experimental settings. While we were unable to evaluate all models or experimental settings, the selected evaluations focus on the most relevant and high-signal conditions. As a result, they provide substantive insight into how different models and human participants understand the intended abstractions of ConceptARC tasks.

Figure 2 shows the results of our rule evaluations on o3, Claude, and Gemini, all using medium-effort + tools, in both textual and visual modalities, as well as evaluations for human-generated rules. Specifically, there are two bars for each model: Correct Grid and Incorrect Grid. The height of each bar corresponds to the percentage of the 480 tasks on which the model's output grid was correct or incorrect. Within each bar, the green section corresponds to tasks for which the model's generated rule was correct-intended; the yellow section corresponds to correct-unintended rules; and the red section corresponds to incorrect rules. The left section of the plot gives these results for the textual modality and the middle section for the visual modality. The rightmost section gives the output accuracy and rule evaluation results for human-generated rules. The gray areas in the human-result bars correspond to human-generated solutions for which we were unable to classify the rule, either because no rule was given by the participant, no rule was collected by the experimenters (this was the case for all of the Incorrect tasks), or the rule given was too unclear to confidently evaluate.

Notably, while o3 in the textual setting rivals humans in output grid accuracy, about 28% of its correct outputs are based on correct-unintended or incorrect rules—indicating reasoning based on superficial patterns rather than intended abstract concepts. We found several types of unintended rules used by models, including rules that described complicated (and spurious) patterns in the demonstrations,

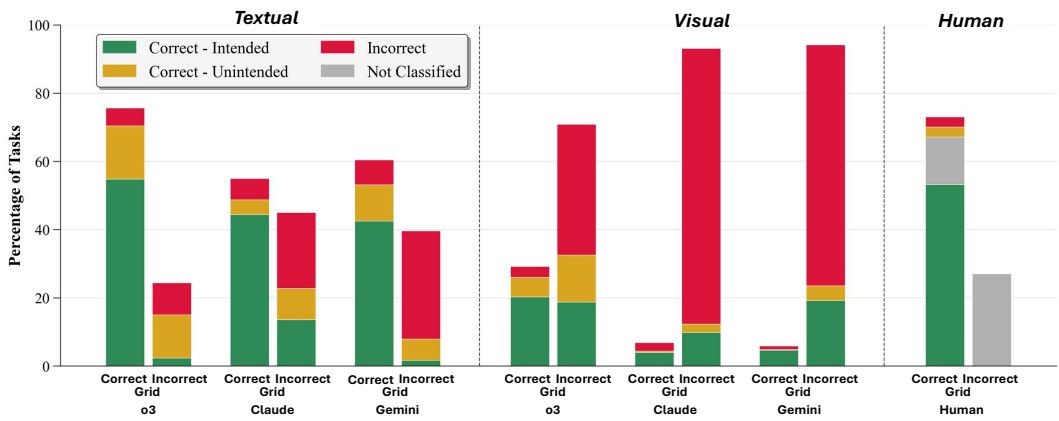

Figure 2: Results of rule evaluations. For each model in each modality (as well as humans), two bars are given, representing the percentage of correct and incorrect grid outputs over the 480 ConceptARC tasks. Each bar shows the fraction of tasks for which the rule is correct-intended, correct-unintended, and incorrect. The gray areas in the human-result bars represent rules that we could not classify—see the section on Rule Evaluation for details. The actual percentages corresponding to regions on the bars are given in Appendix D.

rules that focused on irrelevant features such as (in the textual setting) the specific numbers encoding grid colors, and rules that came close to capturing intended abstractions but included irrelevant spurious associations. Examples of such rules are given in Figure 4. In comparison, we found that only 8% of human's correct outputs were similarly based on correct-unintended or incorrect rules. While our analysis for human-generated rules is limited due to missing rule data (about 20% of rules for correct outputs were not classifiable), this difference is suggestive and should be clarified in future research. Comparing AI models with one another, in cases where Claude and Gemini were accurate on output, both have a smaller fraction of correct-unintended rules than o3, but both are lower than o3 in output accuracy.

Also notable is the percentage of incorrect output grids that are based on correct-intended rules. In these cases, the models recognized the intended abstract rule describing the grid transformation, but were unable to apply it correctly to the test grid. In the textual setting, this seems to be most common in Claude, and less so in Gemini and o3. In the visual setting, however, o3 produced correct-intended rules in about 27% of cases in which its output grid was incorrect; Claude and Gemini did so less frequently, but both still at substantial rates. In summary, looking only at a model's output accuracy in the textual setting—as was done in (Chollet, 2024)—might *overestimate* the model's ability for abstract reasoning, but in the visual domain, accuracy alone might *underestimate* its abstract reasoning abilities. This hints at a direction for improvement in the visual modality across models: models with the capacity to apply the determined rule correctly would be able to substantially improve their output accuracy. These insights illustrate the importance of going beyond simple accuracy in assessing the capabilities of AI models.

While Figure 2 showed our rule evaluations for different models using medium reasoning effort + tools, Figure 3 shows the effects of varying reasoning effort and tool use for the o3 model, in both textual and visual modalities. There are a few important observations to make. First, in the visual setting, increasing reasoning effort from low to medium alone does not have any substantial effect on output accuracy or rule correctness, which aligns with prior work suggesting that test-time scaling does not have the dramatic effects in visual modalities that have been seen in text-only LLMs (Hao et al., 2025). However, enabling Python tool use does result in substantial improvement in output accuracy and rule correctness, especially at medium reasoning effort, likely because the model is able to use computer vision libraries. In contrast, in the textual setting, increasing reasoning effort has a larger positive effect on both output accuracy and rule correctness than enabling Python tool use.

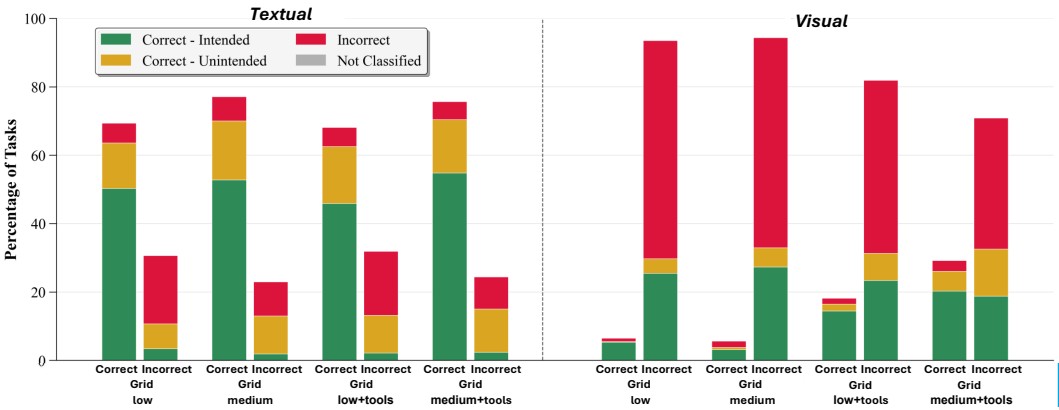

Figure 3: Results of rule evaluations for o3 across all settings. As in Figure 2, two bars showing the percentage of correct and incorrect output grids are included for each setting, with each bar showing the fraction of tasks for which the generated rule is correct-intended, correct-unintended, and incorrect. The actual percentages corresponding to regions on the bars are given in Appendix D.

**Rule-Grid Alignment**    In investigating how accurately the natural-language rules reflect the models' underlying reasoning, our team manually evaluated the solutions generated by o3, Claude Sonnet 4, and Gemini 2.5 Pro in the medium-effort + tools setting for two additional features: rule–grid alignment and visual errors. Rule–grid alignment describes cases where the model's stated rule accurately captures the transformation demonstrated in its output grid. Visual errors refers to cases in visual settings where a grid's incorrectness or inconsistency with its corresponding rule appears to stem from a failure in the model's perceptual apparatus, rather than from a mismatch between its generated rule and its underlying reasoning. Such errors commonly involved failures to recognize the exact grid dimensions, slight inaccuracies in object placement, or incorrect mappings between colors and their numerical encodings.

To perform these evaluations, one team member provided an initial judgment for each task, after which any uncertain cases were discussed as a group until consensus was reached. The results of these evaluations are presented in Table 2. There are two important observations to make from these results. First, the natural-language rules aligned with their corresponding grids in the vast majority of cases; across all evaluated models and settings, agreement exceeded 90% of tasks. This supports the view that the proposed rules generally reflect the reasoning used to produce the grid solutions. Second, the relatively high rate of visual errors corroborates our earlier observations of failure modes in visual settings. Even the best-performing model and setting exhibited a visual error rate approaching 50%, despite maintaining a high degree of rule–grid alignment. This suggests that the low grid accuracies observed in visual settings may be driven in large part by perceptual limitations, rather than solely by differences in reasoning capacity between visual and textual modalities.

Table 2: Percentage of tasks in which different models exhibited either "rule–grid alignment"—the generated rule accurately described the generated output grid—or, in visual settings, a "visual error," where an output appeared to feature some type of perceptual mistake. The "medium effort with tools" setting was used for all models. Percentages are calculated over the tasks in each modality for which the model produced both a valid natural-language rule and a valid output grid.

| Model (medium effort + tools) | Textual | Visual | |
|---|---|---|---|
| | Rule-Grid Alignment | Rule-Grid Alignment | Visual Error |
| **o3** | 98.3 | 97.3 | 49.7 |
| **Claude Sonnet 4** | 91.0 | 96.1 | 60.8 |
| **Gemini 2.5 Pro** | 94.7 | 93.3 | 78.7 |

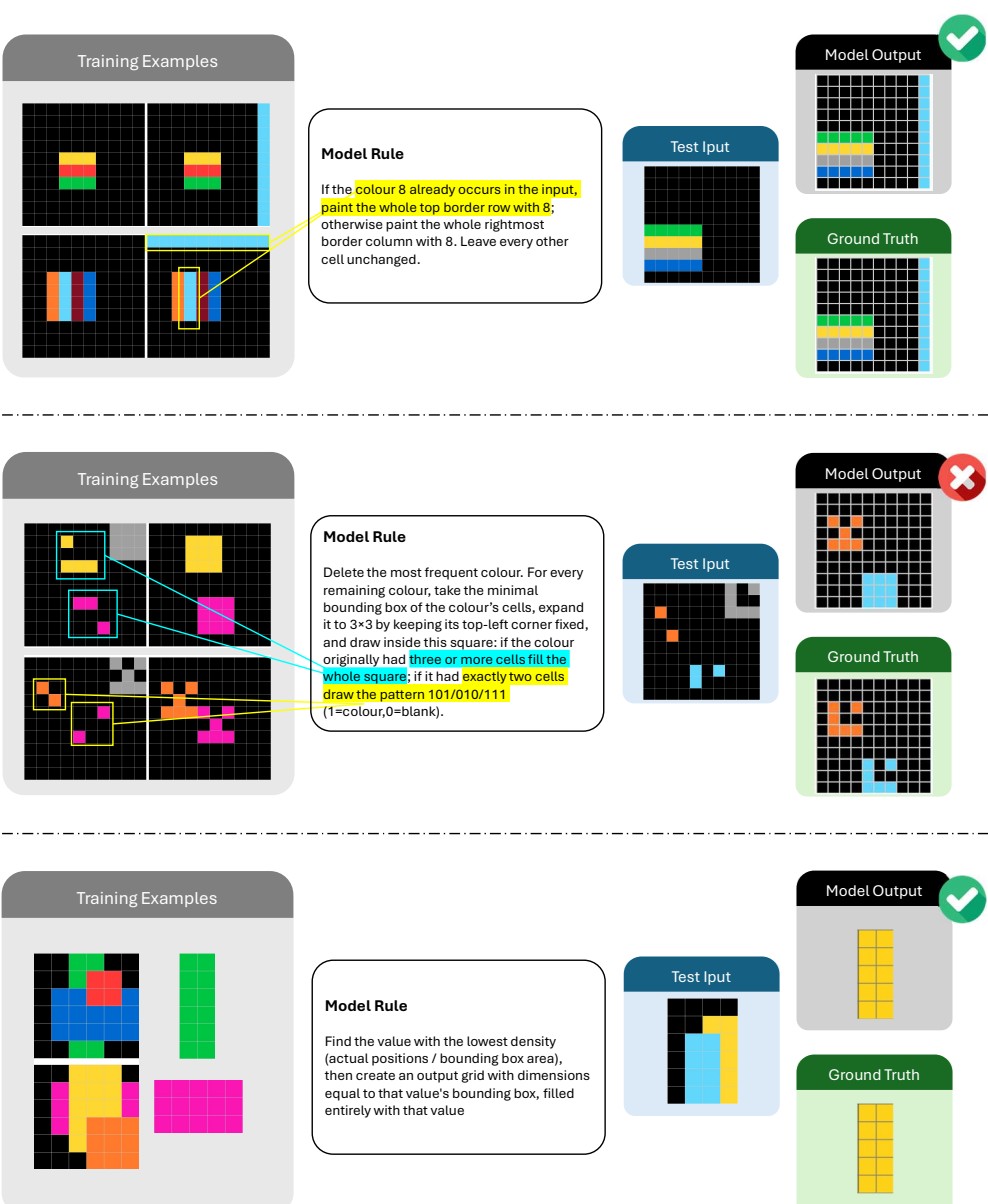

Figure 4: Examples of correct-unintended rules. Top: o3, using medium effort and tools, performs shallow inference for a task from the Horizontal vs. Vertical concept group. The model does not recognize the relation between the orientation of the colored shape components and the blue row, but rather focuses on whether a blue ("8") pixel appears in the grid. In this case, the correct-intended rule works for the given test case, but does not work for other test variants. Middle: o3, using medium effort and tools, on a task from the Complete Shape concept group. The model does not recognize the relation between the colored output shape and the gray prototype and instead overfits to the training examples, producing a correct-intended rule based on shallow features. Bottom: Claude Sonnet 4 uses a density heuristic to approximate the most overlapped figure on a task from the Top vs. bottom 3D group. While this works for some of the test examples, it does not capture the notion of the bottommost shape in a 3D stack, and there are several possible scenarios for which this approach fails.

## 4  DISCUSSION

Given the results described above, we can now provide preliminary answers to the questions we listed at the beginning of this paper. (1) How does the accuracy obtained by AI models compare

with that of humans? Table 1 shows that for textual inputs, o3, with medium reasoning effort, matches or surpasses human accuracy on ConceptARC tasks, with Claude and Gemini obtaining lower accuracy, and o4-mini surpassing humans only when Python tools are enabled. This aligns with results reported in (Chollet et al., 2025; ARC-Prize, 2025).[7] However, using the visual modality, the models' performance still lags significantly behind human accuracy, even when models are given access to Python tools. (2) To what extent do the rules generated by AI models capture the abstractions that were intended by ConceptARC's creators, versus more superficial shortcuts? Figure 2 shows that for textual inputs and medium reasoning effort with Python tools, about 57% of o3's generated rules (regardless of output accuracy) were *correct and intended*; that is, they captured the intended abstractions of the tasks. However, about 28% of o3's generated rules were *correct but unintended*, meaning they were correct with respect to the given demonstrations, and frequently generated correct output grids, but did not capture the intended abstractions. ConceptARC, like ARC, is built on "core knowledge" priors, including "objectness" Chollet (2019), but we found that, for example, o3's rules often focused on colors and individual pixels rather than objects. Moreover, using integers to encode colors enabled unintended shortcuts such as relying on numerical values (e.g., the value for green, 3, is greater than the value for red, 2) which were not available in visual modalities. Both Claude and Gemini's shares of correct-unintended rules (14% and 17% respectively) were lower than o3's, but more than twice the percentage of correct-unintended rules produced by humans (3%). Thus AI models seem more likely to miss intended abstractions and to solve tasks using more superficial features than humans. (3) Regarding the effects of textual vs. visual modalities, Table 1 and Figure 2 show that both output-grid and rule correctness drop dramatically in the visual mode. In addition, we observe that in this mode all three models are considerably better at forming correct-intended rules than generating correct output grids. As for the effects of reasoning effort and Python tools, Table 1 and Figure 3 show that the former is more helpful for textual inputs and the latter is more helpful for visual inputs, especially at higher reasoning effort. These results point to possible directions for strengthening visual reasoning models, especially in more abstract domains.

In short, our results show that models still lag humans in abstract reasoning. Using accuracy alone to evaluate abstract reasoning on ARC-like tasks may overestimate abstract-reasoning capabilities in textual modalities and underestimate it in visual modalities. In evaluating capabilities such as abstract reasoning, our results highlight the importance of going beyond simple accuracy, namely assessing both robustness and the extent to which a system uses generalizable mechanisms rather than more superficial shortcuts (Frank, 2023; Ivanova, 2025; Rane et al., 2025). In order to target these abilities in a meaningful way, we encourage designers of benchmarks and evaluation methods to take the underlying abstractions, as well as derived rules into consideration, in addition to pure output correctness. More generally, developing AI models that grasp the abstractions understood by humans will be essential for these systems to generalize in human-like ways and explain their reasoning in ways understandable to humans—both key abilities for successful human-AI interaction. Our insights further suggest that models do not reliably adopt human priors via training on language and general reasoning. We plan to investigate whether a sole focus on verifying final results does not yield intended abstractions, and whether this issue might be helped by process-based reward models or a more direct inclusion of human-generated reasoning traces. An interesting direction for future research will be to extend our studies to tasks that require more compositional reasoning, such as those in ARC-AGI-2 Chollet et al. (2025).

## 5 RELATED WORK

Several benchmark datasets have been used to evaluate abstract and visual reasoning abilities in LLMs and large reasoning models. Among the ones closest to ARC and ConceptARC are Bongard problems Bongard (1970), letter-string analogies Hofstadter (1985), Raven's progressive matrices (RPMs) Raven (1938), and compositional visual reasoning (CVR) Zerroug et al. (2022). Bongard problems are similar to ConceptARC tasks in that they test understanding of core spatial and semantic concepts, such as "large vs. small" and "inside vs. outside". Like ConceptARC tasks, each Bongard problem is designed to focus on a single such concept. Bongard problems are meant to be solved using visual inputs, and there have been several studies in which multimodal models have been tested on subsets or variations of the original problems; these studies have found that, like our results

---

[7](Kamradt, 2025) noted the large discrepancy between the accuracy of o3-preview (the pre-release version) and the released version of o3 on ARC-AGI-1. The reasons for this discrepancy are not known.

with ConceptARC, the poor performance of VLMs on these problems seem to arise primarily from difficulties with vision rather than with reasoning Małkiński et al. (2024); Pawlonka et al. (2025); Wüst et al. (2024). Letter-string analogies were first proposed by Hofstadter 1985 as an idealized domain for analogy-making. Webb et al. 2023 found that GPT-3 reached human level accuracy 'on letter-string analogies, but other studies (testing both GPT-3 and GPT-4) found that LLMs were not robust to variations of the problems that did not affect humans' performance Hodel & West (2023); Lewis & Mitchell (2025). To our knowledge, no studies have been performed using large reasoning models to solve letter-string analogy problems. Raven's progressive matrices (RPMs) have long been used as tests of human fluid intelligence. RPMs, like Bongard problems, require visual inputs. RAVEN, a dataset consisting of programmatically generated, simplified RPMs Zhang et al. (2019) has been used to evaluate visual reasoning in VLMs (e.g., Zhang et al. (2024); Zhu et al. (2025)), which (as in the case of Bongard problems and ConceptARC) seem to struggle more with perceptual understanding than reasoning. Zerroug et al. 2022 proposed the CVR visual reasoning benchmark, which is meant to test compositional reasoning abilities based on concepts similar to those used in ConceptARC, such as "largest," "inside," "counting," and "contact." They evaluated accuracy on CVR tasks of several convolutional neural networks as well as visual transformers, none of which came close to the abilities of humans when given few training examples. To our knowledge, ours is the only study that evaluates not only the accuracy of models on an abstract reasoning benchmark but also the degree to which the models capture the abstractions intended by the benchmark's designers.

## 6 CONCLUSIONS

The contributions of this work are threefold. (1) We demonstrated the effects of task representation (textual or visual), reasoning effort, and Python tool use on the ConceptARC benchmark for abstract reasoning, finding that in textual modalities with medium reasoning effort, the best AI models match or surpass humans in output accuracy. (2) We evaluated not only accuracy, but also the rules that AI models generated to describe their solutions, and found that while they were able to capture intended abstractions in about half the cases in textual settings, in other cases their rules relied on more superficial features or patterns that are less generalizable. These results suggest that relying on accuracy alone to evaluate abstract reasoning capabilities, as was done in the ARC-Prize challenge, may overestimate the generality of these capabilities. (3) We showed that state-of-the-art multimodal reasoning models still lack human-like visual reasoning abilities, performing dramatically worse in the visual than in the textual modality. However, these models were substantially better at generating correct rules than they were at applying them, which points to directions for improving visual reasoning in such systems. Improving the abstraction capabilities of AI models is an essential direction for future research. Recognizing and using human-like abstract concepts is a crucial step for AI systems to become more generalizable and trustworthy in their reasoning, and also to successfully communicate with humans about their reasoning processes.

## 7 LIMITATIONS

The work we reported here has a number of limitations. Our study involves only the ConceptARC dataset. It is possible that tasks in the original ARC test sets are more resistant to the kinds of rule "shortcuts" seen in our results; however, to our knowledge, there has been no prior research on this topic. Due to resource limitations, we did not experiment with the "high-effort" reasoning setting for o3 or larger reasoning-token budgets for Claude and Gemini—these settings could very well produce significantly more correct-intended rules and higher accuracies. In addition, our classification of human- and machine-generated rules was done manually, and involved some subjectivity; we do not know of any objective or algorithmic means to usefully classify these natural-language rules into our various categories. However, to mitigate individual subjectivity, our team discussed and came to consensus on all potentially ambiguous classifications. Also due to resource limitations, we used pass@1 accuracies for both humans and machines, which differs from the pass@2 and pass@3 accuracies reported in other ARC evaluations. Additionally, we used the same prompt as in the ARC-Prize evaluation of o3 (Chollet, 2024) for the textual setting, and a slightly modified version for the visual setting. It may be that other prompts would elicit better performance for these systems. The data we obtained for human-generated rules was not complete—no rules were collected for incorrect outputs, and even for the correct outputs, in some cases the human-generated rules were not classifiable due to reasons described earlier in the paper.

## ETHICS STATEMENT

This paper uses data from a set of human studies described in Moskvichev et al. (2023). The authors of that study obtained an IRB exemption from the University of New Mexico IRB to perform their study. The data we obtained contained no identifying or other private information about the study participants. We did not identify any other ethics issues with the studies reported here.

## REPRODUCIBILITY STATEMENT

Upon publication we will publish a web page for this paper with all data and code. The ConceptARC dataset is already publicly available at the website `https://github.com/victorvikram/ConceptARC`. The only reasons our results may not be fully reproducible are the non-deterministic nature of the AI models we evaluated, especially due to the fact that our reasoning models were restricted to Temperature 1, and the unpredictability of model releases and deprecations by the companies that created the proprietary models we used. In addition to derived rules and output grids, we collected all reasoning traces by different grids, either directly, or with the most detailed summary setting (OpenAI does not provide reasoning tokens directly). Further, we collected all Python calls by the models, which we interleaved with reasoning data in chronological order, to ensure that model outputs are fully documented.

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

## A  TEXTUAL PROMPT

Find the common rule that maps an input grid to an output grid, given the examples below.

**Example 1**
*Input:*

```
0 0 0 0 0 0 0 0 0 0 0 0
0 0 0 4 4 4 4 0 0 0 0 0
0 0 0 4 4 4 4 0 0 0 0 0
0 0 0 4 4 4 4 0 0 0 0 0
0 0 0 0 0 0 0 0 0 0 0 0
0 0 0 0 0 0 0 0 0 0 0 0
2 2 2 2 2 2 2 2 2 2 2 2
0 0 0 0 0 0 0 0 0 0 0 0
0 0 0 4 4 4 4 0 0 4 4 4
0 0 0 4 4 4 4 0 0 4 4 4
0 0 0 4 4 4 4 0 0 4 4 4
```

*Output:*

```
0 0 0 0 0 0 0 0 0 0 0 0
0 0 0 4 4 4 4 0 0 0 0 0
0 0 0 4 4 4 4 0 0 0 0 0
0 0 0 4 4 4 4 0 0 0 0 0
0 0 0 0 0 0 0 0 0 0 0 0
0 0 0 0 0 0 0 0 0 0 0 0
2 2 2 2 2 2 2 2 2 2 2 2
0 0 0 0 0 0 0 0 0 0 0 0
0 0 0 0 0 0 0 0 0 0 0 0
0 0 0 0 0 0 0 0 0 0 0 0
0 0 0 0 0 0 0 0 0 0 0 0
0 0 0 0 0 0 0 0 0 0 0 0
```

**Example 2**
*Abbreviated*

**Example 3**
*Abbreviated*

> **No Tools Variant**
>
> Below is a test input grid. Predict the corresponding output grid by applying the rule you found. Do not generate any Python code or use any external tools to solve this task.

> **Tools Variant**
>
> Below is a test input grid. Predict the corresponding output grid by applying the rule you found. Use python if needed.

*Test Input:*

```
0 6 6 0 0 6 6 6 0 6
0 6 6 0 0 6 6 6 0 6
1 1 1 0 0 6 6 6 0 0
4 4 4 4 4 4 4 4 4 4
0 0 0 0 0 0 0 0 0 0
0 0 0 0 0 0 0 0 0 0
0 6 6 6 0 0 6 6 6 0
0 6 6 6 0 0 6 6 6 0
0 6 6 6 0 0 0 0 0 0
0 0 0 0 0 0 0 0 0 0
0 0 0 0 0 0 4 4 4 4
0 6 6 6 6 0 0 0 0 0
0 6 6 6 6 0 0 0 0 0
0 6 6 6 6 0 0 0 0 0
```

**Return only this minified JSON (no markdown, no extra keys):**

```
{"rule":"<Transformation rule>","grid":"<final grid>"}
```

## B  VISUAL PROMPT

The left side of the first image shows 3 grids, where each grid square is colored with one of 10 possible colors: black, blue, red, green, yellow, gray, magenta, orange, cyan or brown. The right side of the first image also contains 3 grids, each of which is a transformed version of the corresponding grid on the left. There is a single rule that describes these 3 transformations.

**No Tools Variant**

Determine the rule, and then apply that rule to the grid in the second image. Do not generate any Python code or use any external tools to solve this task.

**Tools Variant**

Determine the rule, and then apply that rule to the grid in the second image. Use python if needed.

You may describe the final grid through natural language using the indices of the different colors, for example:

```
0 0 0 0 0 0
1 1 1 1 1 1
0 0 0 0 0 0
4 4 4 4 4 4
0 0 0 0 0 0
```

which would be a 6x5 grid with a horizontal blue line in row 2 and a horizontal yellow line in row 4. The rest of the grid is black.

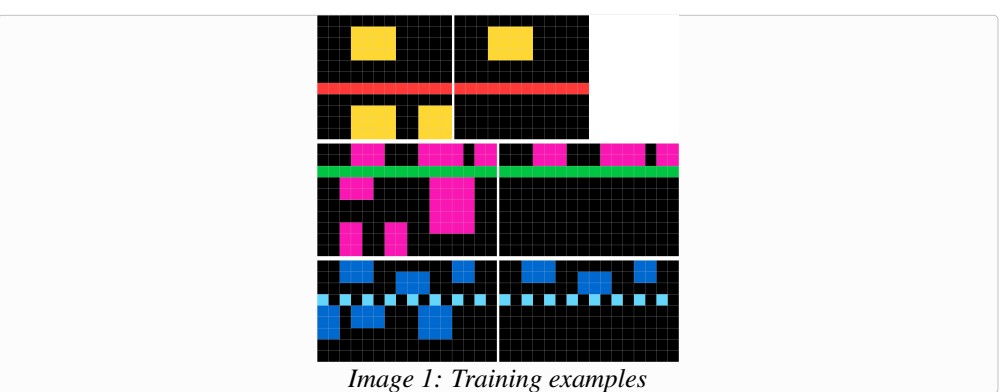

*Image 1: Training examples*

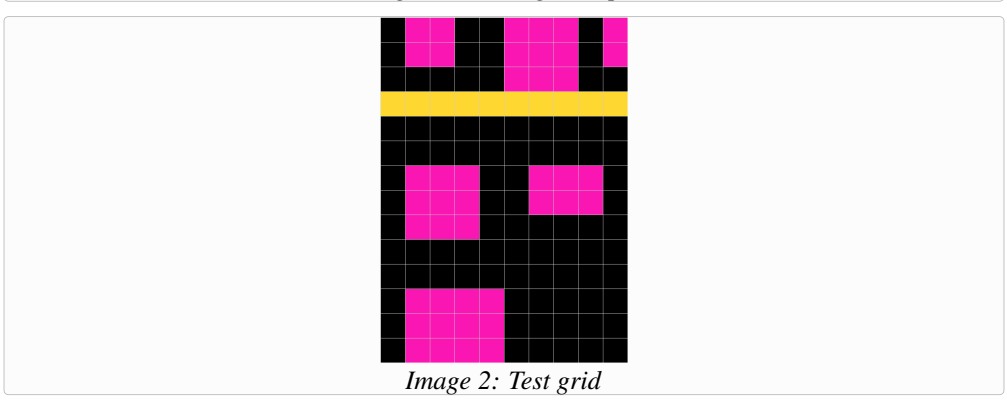

*Image 2: Test grid*

— Return **only** this minified JSON object: "rule":"<Transformation rule>","grid":"<final grid>" No markdown, no extra keys, no code fences. —

## C  PROMPTS FOR NON-REASONING MODELS

The prompts we used for non-Reasoning models were minimally modified to require an additional field containing a reasoning trace in the final JSON object. Otherwise, the prompts were consistent with those used for reasoning models, including variations for visual settings and settings with tools enabled.

## D  DATA FOR RULE EVALUATION PLOTS

Table 3: Data used to create Figure 2. For o3, Claude, Gemini, and human-generated rules, each cell reports the percentage of tasks in a rule classification (Correct-Intended, Correct-Unintended, Incorrect), partitioned by the modality (Textual vs. Visual) and by the correctness of the output grid (Correct Grid vs. Incorrect Grid). Model percentages are computed over 480 total tasks. Human percentages are computed over approximately 4,175 total tests. Rules were not collected for incorrect grids in the original experiment, and so all human responses with incorrect grids are listed here as Not Classified; these percentages are estimates based on the 73% grid accuracy reported by the original experimenters. The final row of statistics for humans show the rule classification breakdown excluding not-classified rules.

| Model (Output Correctness) | Textual | | | Visual | | |
|---|---|---|---|---|---|---|
| | Correct-Intended | Correct-Unintended | Incorrect | Correct-Intended | Correct-Unintended | Incorrect |
| **o3 (Correct Grid)** | 54.8 | 15.6 | 5.2 | 20.2 | 5.8 | 3.1 |
| **o3 (Incorrect Grid)** | 2.3 | 12.7 | 9.4 | 18.8 | 13.8 | 38.3 |
| **Claude Sonnet 4 (Correct Grid)** | 44.4 | 4.4 | 6.3 | 4.0 | 0.4 | 2.5 |
| **Claude Sonnet 4 (Incorrect Grid)** | 13.5 | 9.2 | 22.3 | 9.8 | 2.5 | 80.8 |
| **Gemini 2.5 Pro (Correct Grid)** | 42.5 | 10.6 | 7.3 | 4.6 | 0.2 | 1.0 |
| **Gemini 2.5 Pro (Incorrect Grid)** | 1.7 | 6.3 | 31.7 | 19.2 | 4.4 | 70.6 |

| | Correct-Intended | Correct-Unintended | Incorrect | Not Classified |
|---|---|---|---|---|
| **Human (Correct Grid)** | 53 | 3 | 3 | 14 |
| **Human (Incorrect Grid)** | – | – | – | 27 |
| **Human (Excl. Not-Classified)** | 90.0 | 5.1 | 4.9 | – |

Table 4: Data used to create Figure 3. For all o3 settings, each cell reports the percentage of tasks in a rule classification (Correct-Intended, Correct-Unintended, Incorrect), partitioned by the modality (Textual vs. Visual) and by the correctness of the output grid (Correct Grid vs. Incorrect Grid). All percentages are computed over 480 total tasks.

| o3 Setting (Output Correctness) | Textual | | | Visual | | |
|---|---|---|---|---|---|---|
| | Correct-Intended | Correct-Unintended | Incorrect | Correct-Intended | Correct-Unintended | Incorrect |
| **Low effort (Correct Grid)** | 49.4 | 13.1 | 5.8 | 5.3 | 0.2 | 1.0 |
| **Low effort (Incorrect Grid)** | 4.2 | 7.5 | 20.0 | 25.5 | 4.2 | 63.8 |
| **Medium effort (Correct Grid)** | 52.7 | 17.3 | 7.1 | 3.1 | 0.6 | 1.9 |
| **Medium effort (Incorrect Grid)** | 1.9 | 11.0 | 10.0 | 27.3 | 5.6 | 61.5 |
| **Low effort + tools (Correct Grid)** | 45.8 | 16.5 | 5.6 | 14.4 | 2.1 | 1.7 |
| **Low effort + tools (Incorrect Grid)** | 2.1 | 11.3 | 18.8 | 23.3 | 7.9 | 50.6 |
| **Medium effort + tools (Correct Grid)** | 54.8 | 15.6 | 5.2 | 20.2 | 5.8 | 3.1 |
| **Medium effort + tools (Incorrect Grid)** | 2.3 | 12.7 | 9.4 | 18.8 | 13.8 | 38.3 |

## E  OUTPUT ACCURACY FOR NON-REASONING MODELS

As shown in Table 5 the accuracies of the non-reasoning models were dramatically lower than those of the reasoning models (Table 1). For GPT-4o, in almost all cases in both modalities, the model generated an output grid that was incorrect. For Llama 4 Scout and Qwen 2.5 VL 72B, the same was true in the textual modality; however, for Qwen, in almost all cases in the visual modality, the model was not able to generate an answer at all and did not return the requested JSON format. This was true to a lesser extent for Llama 4 Scout. It is a topic for future research to determine why these models had difficulty generating answers in any valid format.

Table 5: **Non-reasoning models:** Output-grid accuracy (pass@1) on Concept-ARC across models and experimental settings. Accuracy is shown in %. Each cell shows accuracy in the *visual / textual* modality. Temperature is set to 0.0 for all models. Bold numbers correspond to the highest score in each column. The Llama and Qwen interfaces did not provide options for Python tool use.

| Non-Reasoning model | No Python Tools | With Python Tools |
|---|---|---|
| | Textual / Visual | Textual / Visual |
| **GPT-4o** | **14.6** / 0.0 | **8.3 / 0.2** |
| **Llama 4 Scout** | 6.7 / 0.0 | N/A |
| **Qwen 2.5 VL 72B** | 9.2 / 0.0 | N/A |

## F  CONCEPT PERFORMANCE OVERVIEW

ConceptARC (Moskvichev et al., 2023) is organized around 16 basic spatial and semantic concepts. Each concept group consists of 10 tasks that focus around that concept in different ways, testing the derived knowledge on three separate test input grids. Table 6 and Table 7 give the per-concept-group accuracies (each out of 30 grids) of the reasoning models we evaluated (using medium reasoning effort and Python tools), as well as human accuracies on these concept groups from Moskvichev et al. (2023). Humans were tested using visual images of demonstration and test grids, but human accuracy is repeated in both tables for easy comparison.

## F.1 CONCEPT PERFORMANCE COMPARISON FOR TEXTUAL MODALITY

Table 6: **Concept performance (Textual):** Per-concept accuracy (%) on Concept-ARC for medium effort + tools. Best value per concept in bold.

| Concept | Gemini 2.5 Pro | o3 | o4-mini | Claude Sonnet 4 | Human |
|---|---|---|---|---|---|
| AboveBelow | 60 | **90** | 83.3 | 63.3 | 69 |
| Center | 70 | 93.3 | **96.7** | 83.3 | 84 |
| CleanUp | 23.3 | 46.7 | 60 | 46.7 | **89** |
| CompleteShape | 56.7 | **70** | 66.7 | 50 | 71 |
| Copy | 66.7 | 70 | **90** | 56.7 | 78 |
| Count | **86.7** | 80 | 80 | 76.7 | 61 |
| ExtendToBoundary | 60 | **90** | 83.3 | 50 | 81 |
| ExtractObjects | 56.7 | 76.7 | **86.7** | 43.3 | 67 |
| FilledNotFilled | 73.3 | 76.7 | **83.3** | 63.3 | 82 |
| HorizontalVertical | 53.3 | **70** | **70** | 63.3 | 68 |
| InsideOutside | 66.7 | **80** | 73.3 | 43.3 | 68 |
| MoveToBoundary | 63.3 | **80** | 70 | 40 | 78 |
| Order | 50 | 70 | 70 | 40 | **76** |
| SameDifferent | 56.7 | 83.3 | **86.7** | 53.3 | 68 |
| TopBottom2D | 76.7 | 86.7 | **93.3** | 56.7 | 79 |
| TopBottom3D | 46.7 | 53.3 | 56.7 | 50 | **70** |

## F.2 CONCEPT PERFORMANCE COMPARISON FOR VISUAL MODALITY

Table 7: **Concept performance (Visual):** Per-concept accuracy (%) on Concept-ARC for medium effort + tools. Best value per concept in bold.

| Concept | Gemini 2.5 Pro | o3 | o4-mini | Claude Sonnet 4 | Human |
|---|---|---|---|---|---|
| AboveBelow | 0 | 20 | 10 | 0 | **69** |
| Center | 6.7 | 43.3 | 26.7 | 6.7 | **84** |
| CleanUp | 10 | 23.3 | 26.7 | 13.3 | **89** |
| CompleteShape | 3.3 | 30 | 23.3 | 16.7 | **71** |
| Copy | 3.3 | 20 | 23.3 | 3.3 | **78** |
| Count | 16.7 | 53.3 | 50 | 0 | **61** |
| ExtendToBoundary | 0 | 20 | 13.3 | 3.3 | **81** |
| ExtractObjects | 3.3 | 30 | 36.7 | 0 | **67** |
| FilledNotFilled | 6.7 | 30 | 20 | 0 | **82** |
| HorizontalVertical | 3.3 | 33.3 | 20 | 6.7 | **68** |
| InsideOutside | 6.7 | 16.7 | 13.3 | 10 | **68** |
| MoveToBoundary | 3.3 | 30 | 10 | 16.7 | **76** |
| Order | 10 | 33.3 | 36.7 | 13.3 | **60** |
| SameDifferent | 6.7 | 26.7 | 26.7 | 6.7 | **68** |
| TopBottom2D | 13.3 | 33.3 | 50 | 3.3 | **79** |
| TopBottom3D | 0 | 23.3 | 13.3 | 10 | **70** |

## F.3 CONCEPT DIFFICULTY EVALUATION

While, we do not discover a significant correlation between concept difficulty in visual or textual modality, or with human participants, we identify some overarching trends in concept difficulty. We show a full concept performance comparison in the tables Table 6 and Table 7, but specifically point out the performance differences over the concepts "Count" and "CleanUp." We generally compare against the highest performing setting per concept for each performance. Count tasks frequently

involve the production of simple, singular output rows or columns, denoting the count of specific characteristics (e.g shapes, colors, corners). Correspondingly, output grids are often small and easy to generate. In the visual modality, this is the performance closest to humans for both o3 (-7.7%) and Gemini (-44%), and in the textual modality, this concept also results in the biggest positive difference(o3:+32.3%; Gemini:+25.7%; Claude:+15.7%). In contrast, tasks in the CleanUp concept group require the removal of several colors, shapes, or isolated pixels, as well as full reproduction of the remaining input grid. In this concept group, even o3 using medium effort + tools is significantly outperformed by human participants in the visual setting (-65.7%). Similarly, answers to CleanUp tasks constitute the largest negative performance gap in the textual modality (-46.3%). The gap between the other models is even larger. This is a strong indicator that, regardless of modality, models struggle significantly with producing complex output grids.

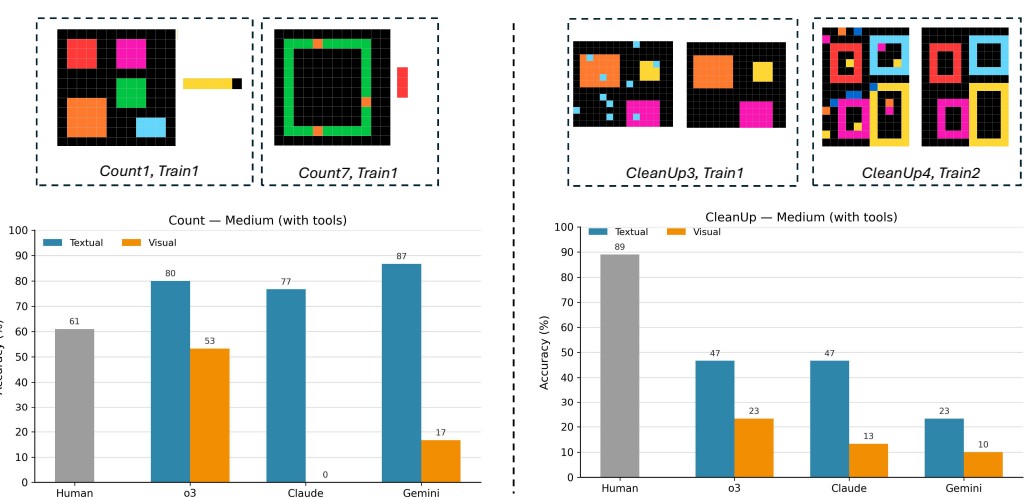

Figure 5: We show two example demonstrations from the concept with the highest and lowest gap between Human and Model performance, CleanUp and Count. Further, we show concept-wise output grid accuracy across three reasoning models in a medium with tools setting (note that we compare against the strongest setting in subsection F.3 instead).

# G CORRECT-INTENDED COVERAGE

Table 8: **Correct-intended task coverage:** Number of tasks covered correctly by category and modality, with coverage rates listed as a percentage of the 480 total ConceptARC tasks. Here, a task is considered "covered" if the model in question produced a correct-intended rule in *any* of its solutions for that task in the given modality. The "AnyModel" rows show task coverage aggregated across all three reasoning models, and the entry for humans shows the coverage of tasks for which at least one human subject produced a correct-intended rule.

| Category | Modality | Covered | Percentage |
|----------|----------|---------|------------|
| Humans | Overall | 475 | 98.96 |
| o3 | Textual | 410 | 85.42 |
| o3 | Visual | 281 | 58.54 |
| Claude | Textual | 343 | 71.46 |
| Claude | Visual | 80 | 16.67 |
| Gemini | Textual | 293 | 61.04 |
| Gemini | Visual | 136 | 28.33 |
| AnyModel | Textual | 451 | 93.96 |
| AnyModel | Visual | 320 | 66.67 |

## G.1 CORRECT-INTENDED COVERAGE IMPLICATIONS

Table 8 clearly shows that, while models (in textual modality) all have a decent coverage, pooling their answers only leads to a moderate increase as compared to the best performing single model (+8%). While the overall coverage is notably lower in visual modality, the increase when pooling the three models again is comparable to textual (+8%). As we do not have individual human performance data, we unfortunately cannot compute similar statistics for pooling single human performances. However, these results again point out stronger abstractive reasoning abilities in a human panel, which only failed to derive the correct abstract transformation in 5 test examples.

## H  ERROR-TYPE OVERVIEW

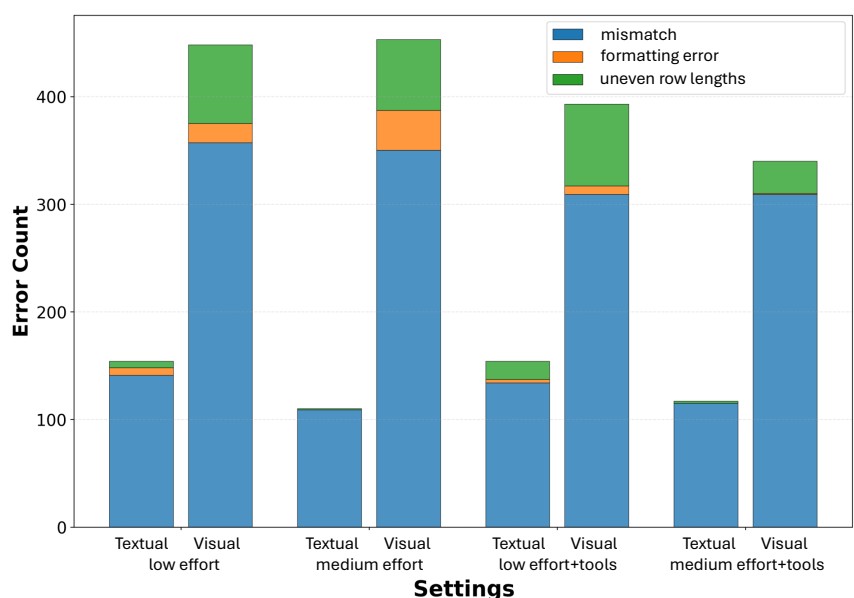

Figure 6: Overview of different error types for o3 in different experimental settings. For each setting, we display textual modality results on the left bar and visual modality on the right bar. The most common error type is a simple mismatch error in which the output grid and the ground truth grid are not identical, including incorrect grid dimensions and single-pixel mismatches. We also encountered some parsing errors, which most often originated from incorrect formatting (see Appendix I). Another parsing error originated from uneven row lengths, which made it impossible to render the candidate grids in the needed rectangle format.

## I  OUTPUT GRID ACCURACIES REASSESSED FOR INCORRECT GRID FORMATS

To compute the accuracies reported in Table 5 and Table 1, we followed the ARC-Prize evaluation method ARC-Prize (2024): we counted an output grid as correct only if it perfectly matched the ground-truth output grid and was in the format requested in the prompt (see Appendix Aand Appendix B). However, upon exhaustive examination of the output grids generated by different models, we found that, in some cases, models generated these answer grids in different formats than that requested in the prompt; these answers were assessed as incorrect. The incorrect output grid formats included surrounding grid rows with brackets, using commas or slashes as row separators, and several other variations.

We re-assessed each case of such formatting to see if the *intended* grid was actually correct. Table 9 gives, for each model and experimental setting, the original output-grid accuracy from Table 1 or Table 5 and the revised output-grid accuracy when incorrect formats are allowed. Table 9 shows that accepting alternate grid formats leads to minor increases in accuracy in most cases, with a few exceptions in which the accuracy rose by more than 5%: o4-mini low-effort, o4-mini low-effort + tools, and Claude Sonnet 4 medium-effort, which had the largest increase: 60.2% to 72.5%.

Figure 7 gives a plot corresponding to Figure 2 but with the revised accuracies. Comparing this to Figure 2, we do not see any substantial changes in the fractions of correct-intended, correct-unintended, and incorrect rules associated with each bar.

In summary, while models sometimes generate their answer grid in a different format than what we requested, whether we accept these formats as valid answers and assess their correctness does not have a large effect on our overall results.

In a smaller number of cases, all in the visual setting, models would generate a natural-language description of the output grid rather than the grid itself. We did not consider these to be in a valid answer format and counted such outputs as incorrect.

Table 9: **Output grid accuracies with alternative grid formats included.** For each model and setting, we give *original accuracy* / *re-assessed accuracy*. Original accuracies are from Table 5 and Table 1.

| Model | Setting | Textual | Visual |
|-------|---------|---------|--------|
| | | Original / Re-assessed | Original / Re-assessed |
| o3 | low effort | 68.3 / 69.4 | 6.5 / 6.5 |
| o3 | medium effort | 77.1 / 77.1 | 5.6 / 5.6 |
| o3 | low effort + tools | 67.9 / 68.1 | 18.2 / 18.2 |
| o3 | medium effort + tools | 75.6 / 75.6 | 29.2 / 29.2 |
| o4-mini | low effort | 52.1 / 59.6 | 3.8 / 3.8 |
| o4-mini | medium effort | 70.8 / 73.8 | 8.1 / 8.1 |
| o4-mini | low effort + tools | 57.3 / 62.5 | 6.7 / 6.7 |
| o4-mini | medium effort + tools | 77.7 / 78.8 | 25.0 / 25.0 |
| Claude Sonnet 4 | medium | 60.2 / 72.5 | 5.2 / 5.2 |
| Claude Sonnet 4 | medium + tools | 55.0 / 59.2 | 6.9 / 6.9 |
| Gemini 2.5 Pro | medium effort | 66.0 / 66.0 | 4.2 / 4.2 |
| Gemini 2.5 Pro | medium effort + tools | 60.4 / 60.4 | 5.8 / 5.8 |
| GPT-4o | No tools | 14.6 / 14.6 | 0.0 / 0.0 |
| GPT-4o | With tools | 8.3 / 13.1 | 0.2 / 0.2 |
| Llama 4 Scout | No tools | 6.7 / 8.5 | 0.0 / 0.0 |
| Qwen 2.5 VL 72B | No tools | 9.2 / 10.0 | 0.0 / 0.0 |

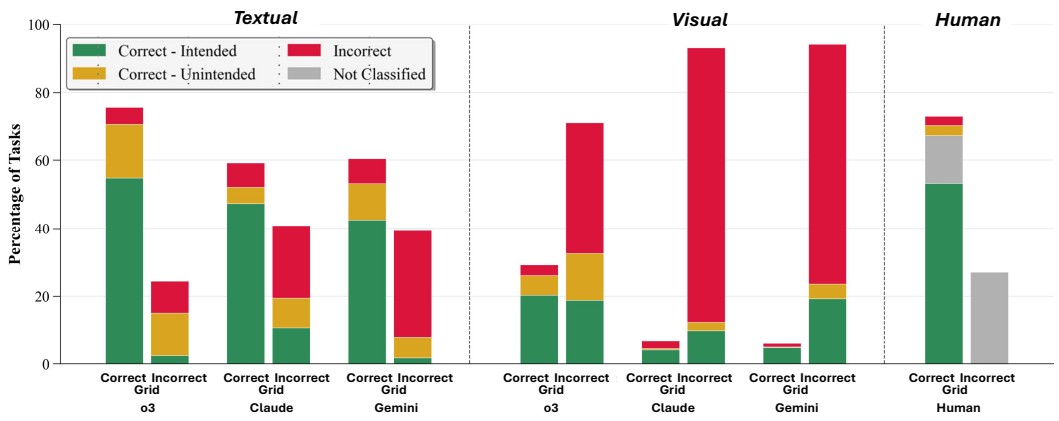

Figure 7: **Re-assessed rule evaluations.** Results of rule evaluations, similar to that shown in Figure 2, but here with re-assessed accuracies.

## J  DISTRIBUTION OF UNINTENDED ABSTRACTIONS ACROSS CONCEPTS

Upon discovering the usage of shortcuts, we were interested in analyzing the distribution of these among different concepts. In particular, models might be systematically employing unintended abstractions on tasks they lack meaningful priors for. Under textual modalities, when models arrived at a "Correct" rule, they produced Correct-Unintended results in about 28.5% those cases with a standard deviation of 16.5%. The concepts with the highest share of unintended abstractions were *TopBottom3D* (70.2%), *CleanUp* (51.2%) and *HorizontalVertical* (42.1%). For the visual modality, the average usage of correct-unintended rules was 26.55%, with a standard deviation of 13.7%. Again, the concepts with the highest share are *TopBottom3D* (60.9%) and *CleanUp* (40%), but also *SameDifferent* (47.83%). *HorizontalVertical* had a reduced share of unintended rules, with only 34%, ranking at fourth-most.

We generally refer to the share of unintended abstractions of correct rules here (intended and unintended), rather than of all rules, in order to account for the difference in rule correctness between modalities. As Table 6 and Table 7 show, *TopBottom3D* is one of the most difficult concepts for models when measured by accuracy (second-lowest in textual, third-lowest in visual), so it is not surprising that they largely rely on unintended abstractions when solving corresponding tasks. Analyzing the proposed rules more closely, few of them actually addressed 3-dimensional arrangement, but instead relied on heuristics, such as density or bounding-box interceptions.

While *CleanUp* produced the lowest overall accuracy in textual modality, it achieved the fourth-highest in visual modality, so it is intuitive to find increased shortcut-usage using text representations. The high share of unintended rules in vision-based inputs is somewhat surprising, but is likely due to difficulties with recognizing global patterns. This is an issue with both modalities, as models tend to employ local, or nested rules, which were overfit to the training demonstrations. In particular, models frequently recognize simple line-based patterns, such as alternating horizontal or vertical lines, but struggle with recognizing separated objects, without clear unifying, global properties besides objectness. In the other named concepts, and on various tasks in general, we recognize several recurring heuristics, including the employment of bounding boxes, four/eight-neighbor connectivity, as well as path finding. These unintended abstractions seemed to be part of a general-purpose tool-box, which models employed for various purposes and not specific to single concepts.

