# OpenReview forum: "Do AI models perform human-like abstract reasoning across modalities?"
_ICLR.cc/2026/Conference — Submitted to ICLR 2026_

### Official Review · Reviewer_RFre · 2025-10-25

**Soundness:** 2
**Presentation:** 3
**Contribution:** 2
**Rating:** 2
**Confidence:** 4

**Summary:**

This paper examines whether high performance on ConceptARC tasks truly reflects human-like abstract reasoning. The authors evaluate several multimodal reasoning models across textual and visual input settings, vary reasoning depth and tool access, and analyze not only output accuracy but also the natural-language rules generated by the models.

**Strengths:**

1. The question addressed is timely and relevant to the ongoing discussion regarding whether LLMs genuinely perform abstraction-based reasoning.
2. The analysis goes beyond output accuracy and attempts to assess conceptual generalization by examining natural-language rule descriptions.
3. The experimental matrix is comprehensive (modalities × reasoning depth × tool access × multiple models).

**Weaknesses:**

1. Although the paper provides a thorough and well-organized empirical analysis, it does not introduce a new method, evaluation metric, or conceptual framework. The contribution is therefore primarily diagnostic and observational, which limits the originality and impact.
2. The conclusions drawn are largely descriptive summaries of observed performance patterns, rather than explanatory insights that deepen our understanding of reasoning mechanisms. The paper stops at “what happens” rather than addressing “why it happens” or “how this informs the design of future reasoning systems”.
3. In visual reasoning, performance may be affected by perceptual encoding (object recognition, segmentation, spatial grouping) as well as abstraction and rule application. Unlike textual inputs, where the perceptual layer is absent, the visual results cannot be directly interpreted as reasoning failures. However, the paper does not experimentally disentangle these two sources of error, making the interpretation of results less reliable.
4. Recent work on reasoning benchmarks has also studied compositional and spatial abstractions. The manuscript lacks a clear articulation of how its findings extend, differ from, or challenge existing conclusions in this line of work. This weakens the contribution framing.

**Questions:**

Referencing the weaknesses, my main concern is about the contribution and novelty claim. The paper provides a thorough in-depth analysis, but it does not appear to introduce a novel method or evaluation approach, and the conclusions read more as descriptive characterizations rather than forming systematic guidance for model or benchmark design.

In addition, for the visual reasoning results, it may be helpful to more explicitly situate the work in the broader visual reasoning literature,  e.g., RAVEN (Zhang et al. 2019), CVR (Zerroug et al. 2022), Bongard problems, where abstract reasoning has been a central theme for many years.

If I have misunderstood the intended contribution or positioning, I would appreciate clarification from the authors.

---

> ### Author Response · Authors · 2025-11-20
> **Author Rebuttal**
>
> We thank the reviewer for insightful comments and for inviting us to clarify their concern with our paper, which we are happy to do.
>
> $\textbf{Paper does not introduce new method, evaluation metric, or conceptual framework, which limits the originality and impact}$
>
> We agree that our contribution is primarily diagnostic in nature; however, we respectfully disagree that this limits the originality or impact of the work. Prior evaluations on ARC-AGI and ConceptARC have overwhelmingly focused on output accuracy, implicitly assuming that a correct output indicates that a model has acquired the intended abstraction.
>
> While some prior work has contrasted transductive versus inductive approaches or analyzed specific transformation rules (Li, et al., 2024), to the best of our knowledge no prior work has systematically examined whether the rules models claim to use actually correspond to the abstract concepts that ARC-style benchmarks were designed to probe. As stated by Chollet (2019) and echoed broadly in the ARC literature, ARC’s fundamental goal is to evaluate core knowledge priors such as objectness, spatial relations, and compositional abstraction, not merely to evaluate pixel-level transformations. Despite this, existing evaluations have not assessed the alignment between intended abstractions and the reasoning processes models employ.
>
> Our work addresses this methodological gap by providing the first systematic evaluation of natural language rules generated by both humans and state-of-the-art multimodal reasoning models across 480 ConceptARC tasks, spanning 16 concepts. We provide evidence that accuracy alone is an unreliable proxy for abstraction capabilities, reveal different failure modes and reasoning shortcuts, and advocate that we should be careful with the interpretation of ARC-style leaderboard claims, particularly those suggesting human-level reasoning or ``breakthroughs".
>
> We believe that introducing this more principled and concept-sensitive evaluation framework constitutes a meaningful new evaluation metric and correspondingly a step forward in the study of abstract reasoning in AI systems.
>
> $\textbf{The conclusions are largely descriptive summaries of observed performance patterns, rather than explanatory insights”.}$
>
> We agree that our current manuscript focuses on describing the mismatch between intended abstractions and the rules models produce, and that we do not yet explore in depth the mechanistic reasons behind these behaviors. We have made this limitation explicit in the revised version under section six.
>
> A key challenge for deeper mechanistic analysis is that the models capable of achieving reasonably high accuracy on ConceptARC (e.g., o3, o4-mini, Gemini, Claude) are proprietary, which prevents us from applying standard explainability tools such as causal interventions or circuit-level methods. As noted in the appendix section E, open-source models with accessible internal structure perform far below human levels, both in accuracy and in rule correctness, so they unfortunately do not provide a meaningful basis for such analyses.
>
> That said, we believe that our findings suggest important insights for future model design, and we have added a dedicated part in our Discussion section with regard to these implications. In particular, our results suggest there may be value of incorporating human reasoning traces during fine-tuning, and of building more fine-grained, abstraction-focused process reward models. Current evaluation schemes typically reward only the final output, which can inadvertently reinforce shortcut-based solutions rather than human-like abstraction. By contrast, process-based rewards or more nuanced validation criteria could help models internalize the intended concepts and encourage the emergence of more robust, human-aligned reasoning priors.
>
>
> $\textbf{Perceptual encoding may impact performance among abstraction and rule application, which is not disentangled, making interpretation unreliable}$
>
> We fully agree that, in the visual modality, errors may arise from perceptual failures in addition to abstraction and rule application.
> To address this concern, we performed an additional round of analysis to disentangle these error sources more explicitly. Building on our existing annotations, we further categorized failures into (1) rule–grid deviations: cases where the model failed to apply the inferred rule correctly, no matter whether the rule is correct or not; and (2) visual errors: cases where the model’s predicted grid structure, object boundaries, or colors were inconsistent with the input vision signal.
>
> We have incorporated this expanded analysis into the revised manuscript Section 3.3 (Rule Grid Alignment) to clarify which portions of the visual performance gap should be attributed to visual perceptual shortcomings versus reasoning challenges. We believe this addition will make our interpretation of visual results more reliable.
>
> [1/2]

---

> ### Author Response · Authors · 2025-11-20
> **Author Rebuttal [2/2]**
>
> [2/2]
>
> $\textbf{Manuscript lacks clear articulation of how its findings extend, differ, or challenge existing conclusions in compositional reasoning benchmarks}$
>
> It is correct that ConceptARC is designed around single abstract concepts rather than fully compositional structures. This does limit what we can say about multi-step or multi-concept reasoning, and we will make this clearer in the revised manuscript.
>
> That said, this focus was intentional. By isolating individual spatial and semantic concepts, ConceptARC lets us examine how well models grasp the core knowledge priors that more complex compositional tasks depend on. Our findings show that models already face challenges in consistently identifying and applying these individual concepts, even in relatively simple settings. This suggests that difficulties seen in compositional benchmarks may arise not only from the composition itself, but also from weaker-than-expected mastery of the underlying abstractions.
>
> We have revised the paper to better clarify how our work relate to tasks requiring compositional reasoning, such as RAVEN, in the "Related Work" section. In particular, many existing reasoning benchmarks, such as Bongard problems, RPMs, letter-string analogies, and CVR, probe abstract or compositional concepts but evaluate models primarily through task accuracy. We emphasize on evaluating not only whether models solve abstraction-centered tasks, but whether the rules they generate actually capture the abstractions intended, which can be harder to diagnose when several concepts are intertwined. Corrspondingly, our work tries to isolate core-concept understanding from the ability to compose multiple concepts, which constitutes a complimentary direction to existing benchmarks on compositional reasoning. In addition, we have included our intention on expanding our work to tasks requiring compositing multiple concepts, such as ARC-AGI2, in Section 4.
>
> $\textbf{Concern regarding contribution and novelty claim. Conclusions read more as descriptive characterizations than forming systematic guidance}$
>
> Our intention with this work is not to propose a new model or algorithm, but rather to challenge an underlying assumption in prior evaluations, namely, that high output accuracy on ARC-style tasks reliably reflects the use of the intended, human-like abstractions these benchmarks were designed to probe.
>
> As mentioned earlier, our work directly fills this gap by providing, to the best of our knowledge, the first systematic evaluation of natural-language rules generated by both humans and state-of-the-art multimodal reasoning models. Prior work has not systematically distinguished between obtaining the correct answer on a task and solving it via the intended abstraction, nor has it evaluated whether model-generated rules align with the conceptual structure of the task. We believe this additional axis of evaluation uncovers an important and previously overlooked dimension of model behavior.
>
> We agree that our initial manuscript could have articulated the broader implications of this perspective more clearly. In the revision, we have highlighted how our findings inform future benchmark design and model development in the Discussion Section. We appreciate the reviewer’s suggestion, which helped us strengthen how we frame the contribution within the larger reasoning literature.
>
> $\textbf{Explicitly situate the work in the broader visual reasoning literature}$
>
> We have added an explicit "Related Work" section to the paper that situates our work in this literature. See Section 5 of our revised manuscript.

---

### Official Review · Reviewer_TPSZ · 2025-10-27

**Soundness:** 3
**Presentation:** 1
**Contribution:** 1
**Rating:** 2
**Confidence:** 3

**Summary:**

This paper examines whether current multimodal reasoning models demonstrate genuine human-like abstraction on the ConceptARC benchmark. The authors evaluate several leading models, including OpenAI’s o3 and o4-mini, Gemini 2.5 Pro, and Claude Sonnet 4, across both textual and visual modalities, varying reasoning effort and the use of external Python tools. Their method introduces a dual-level evaluation framework that measures not only output-grid accuracy but also the alignment of model-generated natural-language rules with the intended abstract concepts of each task. The study finds that while models can match or surpass human accuracy in textual settings, many correct outputs are based on unintended, surface-level patterns. In contrast, visual performance remains much lower, though models sometimes identify correct abstract rules but fail to execute them properly. These findings suggest that accuracy-based benchmarks overestimate abstract reasoning in text and underestimate it in vision, calling for more nuanced evaluation of abstraction-centered reasoning.

**Strengths:**

1. The paper introduces a dual-level evaluation framework that jointly assesses output accuracy and abstraction quality, offering a more reliable measure of reasoning ability.
2. The experiments are comprehensive and well-controlled, providing clear insights into how modality, reasoning effort, and tool use affect abstract reasoning in current multimodal models.

**Weaknesses:**

Although the paper presents a thorough empirical investigation, its contribution remains largely descriptive and methodologically incremental. The analysis primarily focuses on the issue of shortcut reasoning, showing that models often rely on superficial cues such as color indices, pixel-level correlations, or numeric encodings rather than forming generalizable abstract concepts—an issue that has already been well documented in prior research [1]. The authors also mention related phenomena—such as the faithfulness gap between generated natural-language rules and internal reasoning processes, and the execution gap where models correctly identify abstract rules but fail to apply them, however, these observations are discussed qualitatively, without being developed into new analytical frameworks, theoretical explanations, or algorithmic solutions.

The experimental setup is fully based on the existing ConceptARC benchmark, using standard tasks, prompts, and accuracy-based evaluation metrics. The paper does not propose new datasets, task variations, or evaluation protocols that could better isolate or measure abstraction capabilities. As such, it is difficult to view the work as offering a fundamentally new evaluation framework.

Overall, while the experiments are extensive and the findings informative, the paper provides limited methodological novelty and offers relatively few new insights into how to advance abstraction or reasoning in multimodal models.

[1] Samuele Bortolotti et al., A Neuro-Symbolic Benchmark Suite for Concept Quality and Reasoning Shortcuts. NeurIPS 2024.

**Questions:**

How do the authors expect their findings to generalize to other reasoning benchmarks or domains (besides ConceptARC) that involve different types of abstraction (e.g., temporal, numerical reasoning, or others)?

---

> ### Author Response · Authors · 2025-11-20
> **Author Rebuttal**
>
> We thank the reviewer for thoughtful comments on our paper.  Our responses are as follows:
>
> $\textbf{Contribution remains largely descriptive and methodologically incremental. Shortcut reasoning has been documented in (Bortolotti et al. 2024).}$
>
> Indeed, the field has known for a long time that machine learning can lead to models using "shortcuts" or relying on superficial features, and that LLM reasoning models are susceptible to this.  However, we believe that it is important to test for and report on such shortcuts when people are making claims about models attaining (super-)human cognitive capabilities, such as abstract reasoning or understanding of "core knowledge" concepts, especially when these claims are based on performance on widely used benchmarks.  For example, it has been claimed that that the ARC-AGI-1 domain has been essentially "solved" via a breakthrough in LLM's abstract reasoning abilities (Chollet et al., 2025).  Our paper is the only study showing that these claims are over-optimistic. Even though a phenomenon like "shortcut reasoning" has been reported before, it's important (we feel) to continue to publish examples of it in order for the field to judge whether claims of generalizable cognitive capabilities are valid.
>
> We appreciate the reviewer pointing out the Bortolotti et al. paper; however that paper proposes an evaluation method that unfortunately cannot be used to reveal the kinds of shortcuts we discuss in our paper, since it would not only require full access to proprietary models, but also the ability to extract an open-ended set of concepts from those models' latent representations, which is still beyond the capacities of mechanistic interpretability methods.
>
> $\textbf{Mention of faithfulness and execution gap, without development into new theoretical explanations, or algorithmic solutions.}$
>
> Our Response: We have revised the paper to include an analysis of the faithfulness and execution gaps, by quantifying the alignment between generated rules and output grids (see Section 3.3 in the revised manuscript). We do not have theoretical explanations or algorithmic solutions for these gaps (both are very big and open research questions), but we believe that our results are valuable for pointing out directions that research on these topics should take (e.g., faithfulness and execution are quite high in textual settings; perceptual / vision errors are much more common than rule execution errors in visual settings). We have added further insights into the implications of our findings for future model or benchmark development in Section 4 of the revised manuscript.
>
> $\textbf{No proposed datasets, task variations, or evaluation protocols to isolate abstraction capabilities. Difficult to view as new evaluation framework.}$
>
> Our results provide a rough measure of abstraction capabilities, via our classification of correct rules into "correct-intended" and "correct-unintended"---the latter missing the abstractions the tasks are meant to capture. Our analysis of both output accuracy and stated rules has provided the insight that large reasoning models are less capable of humanlike abstract reasoning than has been claimed in the literature. To our knowledge, this kind of analysis has not been performed before to assess results on ARC or related abstract reasoning datasets; instead, prior work has only reported output accuracy, which we show can be a misleading metric of capability. Thus, we would argue that our work does adopt a novel evaluation protocol, which aims at isolating model abstraction capabilities beyond automated verification of outputs.
>
> $\textbf{How do the authors expect their findings to generalize to other reasoning benchmarks or domains that involve different types of abstraction?}$
>
> Our Response: This is an excellent question and will be the subject of future research by our team.  We expect that LLMs and large reasoning models are prone to exploiting shortcuts in many kinds of tasks (e.g., evidence for this in the case of arithmetic reasoning has been reported by Nikankin et al. (2025), Arithmetic Without Algorithms: Language Models Solve Math With a Bag of Heuristics).

---

### Official Review · Reviewer_sHsM · 2025-10-28

**Soundness:** 4
**Presentation:** 4
**Contribution:** 3
**Rating:** 8
**Confidence:** 5

**Summary:**

This paper  investigates the abstraction abilities of large SOTA reasoning AI models (o3, o4-mini, Google’s Gemini 2.5 Pro,  Anthropic’s Claude Sonnet 4) using the ConceptARC benchmark. They show that the  (1) using accuracy alone to evaluate human-like abstract reasoning is insufficient, due to the frequent use of shortcuts by the modes, and (2) analysis of the models' linguistic explanations of their solutions reveal that models often fall short of using human-like abstractions intended by ConceptARC design, even if the final output is correct.

**Strengths:**

The paper is well written and well grounded in cognitive theory.

Theoretically sound experiment design.

This work is a broadly interesting, novel, and theoretically valuable contribution.

**Weaknesses:**

The authors acknowledge that the study still relies on the natural language rule being a faithful description of the internal procedure.

**Questions:**

1. It would be informative to understand failure modes, as they were being classified by the authors. For instance the authors note that (1) "o3’s rules often focused on colours and individual pixels rather than objects" and relying on integer number for colours, such as (2) "value for green, 3, is greater than the value for red, 2".  I'm not convinced that  (2) should be treated as a shortcut, because a human given the same puzzle, in text form only, could conceivably reason about number is the same way.

2. Have the authors considered analyzing the models' reasoning in forms other than the text explanations, but perhaps including also the Python code that was used by the models, or asking the models to generate programmatic implementations of their solutions?

---

> ### Author Response · Authors · 2025-11-20
> **Author Rebuttal**
>
> We thank the reviewer for their positive comments and suggestions for our paper.
>
> $\textbf{The study still relies on the natural language rule being a faithful description of the internal procedure.}$
>
> In a new analysis, we were able to manually verify the faithfulness of the rules generated by o3, Claude Sonnet 4, and Gemini 2.5 Pro in their "medium effort with tools" settings. Similarly to our rule evaluations, we manually inspected the proposed rule and the output grid and assessed the degree of alignment between the two. The new results are discussed thoroughly in Section 3.3 (Rule-Grid Alignment) of our revised manuscript; in short: our results indicate that natural-language rules are aligned with their generated output grids in the vast majority of cases, with alignment rates of over 90\% in all models across both modalities. We believe that these insights further strengthen our claim regarding the usage of unintended abstractions.
>
> $\textbf{It would be informative to understand failure modes, as they were being classified by the authors.}$
>
> Unfortunately we were unable to include a detailed examination of all failure modes, as there are many ways in which LLMs can use superficial features in their solutions. We do analyze error types at a higher level, but a more detailed analysis would require an in-depth consideration of underlying reasoning traces, program calls, and the development of a new classification scheme. The reviewer might, however, find the new results on concept-level usage of shortcuts we added in appendix (J) interesting. While we did not go into detail about different failure types, we briefly discuss the usage of concept-specific heuristics (density, bounding-box overlaps...). Nevertheless, we agree that an in-depth analysis of failure modes would be an interesting direction for future work.
>
> (2) We agree that human test-takers, if presented with a textual version of a task, might also make use of text-based features, such as the numerical representation of colors. One of our central motivations was precisely to examine how modality influences the use of human-like abstractions. However, this does not change the ARC benchmark’s core intended aim—to assess the capacity for human-like abstraction and the degree to which the solver possesses the appropriate conceptual priors. Accuracy on ARC and related benchmarks is often explicitly interpreted as a proxy for reasoning ability, and the models' reliance on textual features calls this view into question: if models achieve high accuracy by leveraging representational shortcuts unavailable or infeasible to humans solvers using the typical format, it challenges the benchmark’s intended interpretability. Since standard language-model evaluation on ARC tasks uses this same textual format, our findings from ConceptARC suggest that their observed performance may not faithfully capture the reasoning abilities the benchmark was designed to measure.
>
> $\textbf{Have the authors considered analyzing the models' reasoning in forms other than the text explanations?}$
>
> Though this is an interesting direction for future research, it is out of the scope of the present work. Although LLMs frequently utilized Python programs to form their solutions in our tool-enabled experiments, we did not conduct a detailed analysis of how these programs aligned with the models’ own natural-language reasoning. Reviewing and interpreting the large number of code snippets for each task was infeasible given our current resource constraints. Thus, our focus here is on analyzing natural-language solutions to gain insight into LLMs’ reasoning processes, and, in particular, the extent to which performance accuracy on ARC and related benchmarks reliably reflects abstraction and reasoning ability. However, there has been prior work on this topic, which investigated how LLMs can compose atomic concepts, represented as programs, to solve complex and highly compositional tasks in both ARC and ConceptARC (Li et al., Combining Induction and Transduction for Abstract Reasoning, 2024). The use of induction to generate a program solution, which in turn generates the output grid, is an interesting alternative to natural language, but poses difficulty in clearly identifying the priors used to solve the task.

---

> > ### Comment · Reviewer_sHsM · 2025-11-24
> > **Thank you for the response.**
> >
> > I retain my score recommending acceptance.

---

### Official Review · Reviewer_CGz5 · 2025-10-30

**Soundness:** 2
**Presentation:** 1
**Contribution:** 1
**Rating:** 0
**Confidence:** 4

**Summary:**

The paper audits LLMs/VLMs on ConceptARC, asking whether model-generated “rules” reflect intended abstractions or shortcut heuristics. It compares text vs. vision settings, with/without external tools, and uses human labels to categorize rules (intended vs. “correct-unintended”). Main takeaways: text looks relatively strong but often rides shortcuts; vision lags largely due to execution/Per-pixel perception; tools and reflective feedback help somewhat.

**Strengths:**

1. Full human evaluation on model-produced solutions; appreciate the effort.

2. Overall, easy to read.

**Weaknesses:**

1. **Missing literature review / comparison to baselines**, which makes it hard to position the work—especially given the ARC-AGI motivation vs. ConceptARC-only experiments (ConceptARC is generally easier: smaller search space, limited OOD).


2. **Novelty is limited.** While this paper is fully empirical, the main insights are already well-known in the community: LLMs struggle on complex multi-hop tasks (e.g., ARC-AGI2), models can be poor executors even with a correct NL instruction, VLM per-pixel recognition is a bottleneck, and feedback from an external executor (“Reflection”) helps.


3. I disagree with the “correct-unintended” labeling example (lines 188–190). If a rule solves all demos, it should be treated as an alternative intended rule unless the demos explicitly disambiguate competing hypotheses. Priors are mentioned but not given for either ARC-AGI or ConceptARC about how to craft solutions; moreover, “per-pixel objectization” can be a valid abstraction in this paradigm. Please provide a sharper operational definition: e.g., classify as correct-unintended only when a rule fits all train demos but fails the held-out test specifically because the demos did not disambiguate (As in Figure 4 Middle)—under this criterion, the lines 188–190 example would not qualify since it also solves the test.


As written, the work lacks publishable research contribution or scholarly form: related work and baselines are missing, and the novelty is limited.

**Questions:**

1. Given the identical setup, why not test on ARC-AGI or ARC-AGI2?

Minor:
1. Figure 2 legend box has dots overlaying text.

---

> ### Author Response · Authors · 2025-11-20
> **Author Rebuttal**
>
> We thank the reviewer for their thoughtful comments on our paper.  We respond to these points below.
>
> $\textbf{Missing literature review / comparison to baselines}$
>
> In our initial manuscript, we included a brief overview of relevant research in our introduction, but agree that it will strengthen the paper to address relevant research more explicitly. We have added such a section (see Section 5 of the revised pdf).  Regarding the missing comparison to existing baselines, we would like to note that we are discussing a novel evaluation approach, which, to the best of our knowledge, has not been discussed on benchmarks in the ARC domain in prior work. If the reviewer has specific baselines in mind that would strengthen the paper, we would be happy to consider adding them to this study.
>
> $\textbf{Choice between ConceptARC and ARC unclear}$
>
> As we stated in our initial manuscript in lines 94-96, we choose ConceptARC to isolate simple, verifiable abstractions. We agree with the reviewer that ConceptARC is generally simpler than ARC-AGI, which we believe to be an additional argument for the choice of ConceptARC. The usage of unintended abstractions on these simple, isolated tasks constitute a useful baseline.  The simplicity of these tasks allows us to fairly easily make sense of LLM-generated rules. Tasks that involve composing multiple concepts, as required in ARC-AGI1, and especially ARC-AGI2, are more complex to analyze, and may result in an even higher utilization of shortcuts. We have clarified this choice in our revised manuscript within the introduction section.
>
> $\textbf{Novelty is limited. the main insights are already well-known in the community.}$
>
> We agree with the reviewer that the struggle of LLMs on multi-hop reasoning tasks, including those requiring composition of multiple concepts or abstractions, are a well-known challenge.  What's novel about our study is that it counters claims in the literature that large reasoning models (like o3) have essentially "solved" the kind of abstract reasoning needed in the ARC-AGI-1 domain (e.g., Chollet et al., "Arc-AGI-2: A new challenge for frontier AI reasoning systems". It is interesting that ConceptARC tasks typically do not require multi-hop reasoning, which is one of the reasons for why we opted for ConceptARC instead of ARC-AGI(2), and even on that dataset, we find that reasoning models still rely on shortcuts on a significant fraction of tasks.
>
> As for models being poor executors, we aim to point out the difference between failing to execute a correct rule (or an incorrect rule), and actually failing to understand the transformation on a conceptual level. We find that for the models we tested, the output grids tended to be faithful outcomes of the generated rule, whether it was correct or not (see Table 2 and section 3.3 in the revised pdf). Thus these models were good at executing rules, but not as good at forming rules that captured the intended abstractions.
>
> Further, we appreciate that VLM pixel-recognition is a well-known bottleneck. Our results indicate that VLMs may indeed struggle with pixel recognition, but interestingly, models can still infer the correct transformation rule in up to 30\% of cases in some settings.  This is a 3x increase compared to the performance estimation of pure output-based evaluation. We hope that the reviewer can appreciate this as a meaningful insight.
>
> [1/2]

---

> ### Author Response · Authors · 2025-11-20
> **Author Rebuttal [2/2]**
>
> [2/2]
>
> In general, we would like to clarify our perspective on the Abstraction and Reasoning Corpus as a benchmark of human-like reasoning. Recent papers claim that ARC is "solved", and correspondingly see a huge leap on abstract reasoning abilities in Language Models [1]. As Chollet states in [2]: "we present [...] the Abstraction and Reasoning Corpus (ARC), built upon an explicit set of priors designed to be as close as possible to innate human priors.", ARC, as well as ConceptARC are designed to elicit human-like priors. In our paper, we do not merely argue that performances have been overestimated, but we propose that prior evaluation may ignore this crucial part. The limitations the reviewer lists are indeed well-known, while we believe that, even without this additional perspective, our insights on them are a meaningful contribution, they are not our sole focus. Our paper empirically shows that there is a disconnect between the pure output-centric evaluation used for the aforementioned claims, and the evaluation of the actual usage of human core knowledge priors. We believe that it is imperative for the advancement of the science of AI to do proper evaluation for a claimed target, especially on claims of cognitive capacities like abstraction, and this is one of the main research contributions of our paper.
>
> [1] Chollet, F., Knoop, M., Kamradt, G., Landers, B., & Pinkard, H.  ARC-AGI-2: A new challenge for frontier AI reasoning systems. arXiv preprint arXiv:2505.11831, 2025.
> [2]: Chollet, François. On the Measure of Intelligence. arXiv:1911.01547, 2019.
>
> $\textbf{"I disagree with the “correct-unintended” labeling example (lines 188–190). Priors are mentioned but not given."}$
>
> To clarify, the notion of "Correct-Intended" is the set of intended, human-like abstractions, which is what the ARC (and ConceptARC) benchmarks have been claimed to test [2]. We would argue that any set of demonstrations allow for ambiguous rules; in general it is almost impossible to fully disambiguate.  However, crucially, humans are still able to recover the intended abstractions due to their shared priors, even when they are not shared explicitly. Given our results on evaluating human rules, we are able to confirm this empirically.
> The reviewer is right that any rule that describes the demonstrations, however non-humanlike, is technically correct, and we do refer to them as such. However, we further distinguish based on the notion of intended abstractions, in order to showcase a remaining difference compared to the solutions produced by humans. As we argue in the paper, human-aligned concepts will be essential for many types of AI-human teamwork, and for AI systems to be trustworthy in the human world.
>
> $\textbf{Why not test on ARC-AGI1 or ARC-AGI2?}$
>
> The choice of ConceptARC is indeed an intentional one, which we outlined in our prior response, as well as in our new manuscript version. However, we certainly agree with the reviewer that evaluation on ARC-AGI1 or ARC-AGI2, as well as other benchmarks for abstract reasoning, is an important future step.
>
> $\textbf{Dotted Overlay in Figure 2}$
>
> We thank the reviewer for bringing this to our attention and corrected the figure in our revised manuscript.

---

> > ### Comment · Reviewer_CGz5 · 2025-11-26
> >
> > Thank you for the revision. I still find that the new Related Work section mostly surveys other benchmarks (Bongard, RPM, CVR) but omits closely related LLM-on-ARC studies that both run LLMs on ARC/ConceptARC and analyze their reasoning processes. This is not a benchmark paper. Some highly related papers to consider are (non-exhaustively):
> >
> > * Acquaviva et al., *LARC: Language Annotated Abstraction and Reasoning Corpus* (CogSci 2021).
> > * Opiełka et al., *Do Large Language Models Solve ARC Visual Analogies Like People Do?* (CogSci 2024).
> > * Moskvichev et al., *The ConceptARC Benchmark: Evaluating Understanding and Generalization in the ARC Domain* (arXiv 2023), which already evaluates GPT-4 on ConceptARC.
> > * Xu et al., *LLMs and the Abstraction and Reasoning Corpus: Successes, Failures, and the Importance of Object-based Representations* (TMLR 2024).
> > * Tan, *An Approach to Solving the Abstraction and Reasoning Corpus (ARC) Challenge* (arXiv 2023).
> > * Tan & Motani, *Large Language Model as a System of Multiple Expert Agents: An Approach to Solve the ARC Challenge* (CAI 2024).
> >
> > Given this existing body of work, the statement that *“ours is the only study that evaluates not only the accuracy of models on an abstract reasoning benchmark but also the degree to which the models capture the abstractions intended by the benchmark’s designers”* is overstated and should be toned down and situated relative to prior LLM–ARC evaluations.
> >
> > Overall, this paper is about empirically evaluating ConceptARC with LLMs/VLMs and providing insights, but in its current form it still lacks novelty, clear positioning, and the level of methodological and scholarly rigor expected at ICLR. I appreciate the authors’ effort, but my score remains 0.

---

> > > ### Author Response · Authors · 2025-12-01
> > > **Authors' response to Reviewer CGz5**
> > >
> > > We respectfully disagree with the reviewer that the paper lacks novelty, clear positioning, and rigor. We stand by our statement in the paper that, to our knowledge, “ours is the only study that evaluates not only the accuracy of models on an abstract reasoning benchmark but also the degree to which the models capture the abstractions intended by the benchmark’s designers.”
> > >
> > > There is a large literature on ARC, which includes several hundred papers; we limited our discussion of related work on ARC to papers directly related to our topic: those that discuss what the ARC domain is meant to evaluate, as well as reports on the performance of “reasoning models” like o3 on ARC.  Such citations were included throughout our paper, and in our related work section we surveyed work on related benchmarks and attempts to use them to evaluate abstract reasoning in LLMs.
> > >
> > >  The reviewer lists six papers that they claim “are closely related LLM-on-ARC studies that both run LLMs on ARC/ConceptARC and analyze their reasoning processes.”  **However, this is incorrect: none of these papers analyze the reasoning process of LLMs in any deep and systematic way** (and notably, all of these papers were published prior to the release of o3 and related reasoning models).  **Unlike our work, none of these papers explores the extent to which AI models' accurate outputs on ARC-like tasks are based on the intended core-knowledge abstractions that ARC was designed to evaluate.**
> > >
> > >  Below we provide a brief review of each paper listed by the reviewer, and highlight how they differ from—and do not address—the core contributions of our works.
> > >
> > >  The reviewer did not respond to our other responses to their queries.
> > >
> > > *Acquaviva et al., LARC: Language Annotated Abstraction and Reasoning Corpus (CogSci 2021).*  This was published as an abstract in Cogsci2021, and expanded into a longer paper, Communicating Natural Programs to Humans and Machine, NeurIPS 2022.  The paper describes the LARC dataset, in which a subset of the ARC training problems is annotated with human-generated instructions for transforming the test grid.  The authors used a pre-trained, fine-tuned LLM, with and without the human-generated instructions, to solve a subset of ARC tasks, finding that the instructions only slightly improve the (rather low) base accuracy.  **The authors only reported accuracy and did not include any analysis of reasoning or the rules the model used to form correct answers.**
> > >
> > > *Opielka et al., Do Large Language Models Solve ARC Visual Analogies Like People Do? (CogSci 2024).*  This paper compares LLMs and humans (children and adults) on a very limited dataset: namely, two 8-task sets of very easy ARC-like tasks. The authors look at the frequency of four types of errors made by each group.  **In contrast to our study on 480 tasks, this study does not analyze the reasoning behind how correct answers are obtained.**
> > >
> > > *Moskvichev et al., The ConceptARC Benchmark: Evaluating Understanding and Generalization in the ARC Domain (TMLR 2023).* This paper (which we do cite in our paper) presents the ConceptARC data set, and compares humans and GPT-4 (using textual inputs) on the 480 tasks.  (For our paper, we were able to obtain and analyze previously unpublished data from their human study.)  They show that GPT-4 obtains relatively poor accuracy on the ConceptARC, but, **unlike our paper, they do not do any analysis of reasoning done by GPT-4 or humans to obtain their answers.**
> > >
> > > *Xu et al., LLMs and the Abstraction and Reasoning Corpus: Successes, Failures, and the Importance of Object-based Representations (TMLR 2024).* Using 50 tasks from the ARC training set, the authors show that providing GPT-4 with a more structured, object-based representation of the tasks improves performance.  The authors use linear regression to find relationships between success and number of pixels and colors in a task.  The authors also looked at CoT output provided by the model in 13 tasks that were correctly solved by GPT-4 and found only three tasks were accompanied by the "correct reasoning steps."  **There was no discussion on how they determined "correctness" of reasoning steps, or whether the reasoning steps given were faithful.**
> > >
> > > *Tan, An Approach to Solving the Abstraction and Reasoning Corpus (ARC) Challenge (arXiv 2023).*  This paper is unpublished, and examines only a small number of simplified ARC tasks. The authors use a simplified version of ARC with maximum grid size 8x8. The paper seems to evaluate LLMs on only 4 tasks.  The authors test the effects of providing “human priors” in the prompt, with instructions on how to identify "objects" and a list of the possible grid transformations.  While the prompt asks the model to provide a rule, **the paper contains no analysis of the resulting rules or any other aspect of reasoning.**
> > >
> > > (continues in next box)

---

> > > > ### Author Response · Authors · 2025-12-01
> > > > **Continuation of authors' response to Reviewer CGz5**
> > > >
> > > > *Tan & Motani, Large Language Model as a System of Multiple Expert Agents: An Approach to Solve the ARC Challenge (CAI 2024).*  This paper uses GPT-4 on a subset of the ARC training tasks, looking at the effects of providing the model with different “abstraction spaces,” that is, textual inputs that include pre-segmented objects or the explicit coordinates of each colored cell in the grid.  The extra information provided to the LLM makes this an entirely different task than that proposed by Chollet.  **The model is asked to describe the simplest input-output relationship for all input-output pairs, but the authors did no analysis on the results.**

---

### Official Review · Reviewer_abxG · 2025-11-01

**Soundness:** 3
**Presentation:** 3
**Contribution:** 2
**Rating:** 6
**Confidence:** 3

**Summary:**

This paper presents a rigorous and thoughtful empirical investigation into how state-of-the-art multimodal reasoning models perform on the **ConceptARC** benchmark. Unlike previous evaluations that focused solely on grid-level accuracy in ARC-like problems, the authors analyze **whether models capture intended abstractions**, by jointly assessing (i) grid accuracy and (ii) natural-language rules generated by the models to explain their reasoning.

The study spans **textual and visual modalities**, **varying reasoning budgets**, and **tool-use settings** (with or without Python execution). The dual evaluation provides new insight into how much of models’ apparent success arises from genuine abstraction versus “shortcut” pattern matching. Key findings include:

* Models such as o3 match or surpass human accuracy in textual settings, but perform far worse visually.
* Even when textual outputs are correct, nearly 30% of o3’s reasoning rules reflect unintended or spurious abstractions.
* In the visual modality, models often produce correct-intended rules but fail to apply them accurately—indicating procedural rather than conceptual limitations.
* Allowing Python tool use substantially improves performance in visual settings, likely by leveraging external computer vision functions.

**Strengths:**

* **Timely and meaningful contribution.** The work directly addresses the gap between performance-based and concept-based measures of intelligence, an issue at the heart of ARC/ARC-AGI.
* **Methodological novelty.** The dual-layer evaluation of grid accuracy and rule abstraction is methodologically elegant and exposes nuanced differences between human and model cognition.
* **Comprehensive experimental design.** The study systematically controls for modality, reasoning effort, and tool access across four reasoning models and three baselines.
* **Rich interpretability and cognitive framing.** The authors’ use of “intended vs. unintended rules” operationalizes abstraction fidelity in a manner that is both cognitively interpretable and empirically testable.
* **Human-model comparison adds depth.** By aligning human rule data with model-generated rules, the paper quantifies abstraction alignment rather than relying solely on raw success rates.

**Weaknesses:**

* **Limited interpretability of rule faithfulness.** While natural-language rules correlate with generated outputs, there is no quantitative measure of *rule–behavior alignment*. This weakens the claim that models “used” those abstractions.
* **Subjectivity in annotation.** Rule classification (correct-intended / unintended / incorrect) relies on manual consensus; inter-rater reliability or annotation protocol statistics are missing.
* **Incomplete exploration of reasoning-effort scaling.** High-effort o3 and extended token budgets were omitted due to cost, leaving open whether abstraction quality scales further.
* **Under-analyzed abstraction bottleneck.** The paper identifies “shortcut” reasoning but offers little mechanistic or representational explanation (e.g., which components cause pixel- or color-based shortcuts).

**Questions:**

Please see the weakness above.
1. Do you encounter issues with annotation consistency or ambiguity in classifying rules, and how was inter-rater agreement verified?
2. How sensitive are the results to prompt design, especially in the *visual* modality where slight wording changes might affect spatial parsing?
3. Have you examined whether certain concept groups (e.g., *CleanUp*, *ExtendToBoundary*) systematically elicit shortcut reasoning, suggesting task-type bias?

---

> ### Author Response · Authors · 2025-11-20
> **Author Rebuttal**
>
> We appreciate the reviewer's positive comments as well as pointing out weaknesses of our paper.
>
> $\textbf{Limited interpretability of rule faithfulness.}$
>
> We have manually analyzed the faithfulness of the rules generated by o3, Claude Sonnet 4, and Gemini 2.5 Pro for each of the 480 tasks in the "medium effort with tools" setting for both textual and visual input. That is, we assessed whether the model's generated output grid aligns with the rule it gave. This analysis was generally straightforward, but our full team discussed any potentially ambiguous classifications. We have included the results of this analysis in the revised version of our manuscript in Section 3.3 (Rule-Grid Alignment). To give a short overview, models are aligned with their proposed rules at rates consistently over 90\%, regardless of modality or of rule or output-grid correctness.  This provides evidence that here the models' generated rules are mostly faithful to their reasoning process.
>
> $\textbf{Subjectivity in annotation:  Rule classification relies on manual consensus; inter-rater reliability or annotation protocol statistics are missing.}$
>
> We have added a detailed description of our rule-classification protocol to the revised paper at the beginning of the section on rule evaluation. In short: A subset of four members of our team performed all the rule classifications. To create the data for Tables 2 and 3, each rule was first classified by one of these members, and then reviewed by a second member. When these team members were uncertain or in disagreement about a classification, the rule was discussed together by all four team members until a consensus was reached. Upon publication of the paper, all classifications will be released on the paper's publicly available website, so anyone can examine our classifications.
>
> $\textbf{Incomplete exploration of reasoning-effort scaling.}$
>
> Unfortunately we were limited in time and funding, so were not able to test a larger number of models or higher reasoning budgets. It is likely that output-grid accuracy would increase with higher reasoning budgets, as it does in tests of ARC-AGI-1 (+12\% for medium and +6\% for high respectively; https://arcprize.org/leaderboard). We do not know whether abstraction quality will scale in the same way.  Figure 3 in our paper shows that the ratio of correct-intended to correct-unintended rules does not change substantially when moving from low to medium effort, in fact, correct-unintended percentage shows a small increase with medium effort over low effort. Whether this remains the case for higher reasoning budgets remains a topic for future research.
>
> $\textbf{Under-analyzed abstraction bottleneck: The paper identifies “shortcut” reasoning but offers little mechanistic or representational explanation.}$
>
> We agree that mechanistic insights into the underlying cause for the usage of shortcuts would be very interesting. Unfortunately, a mechanistic or representational investigation is not possible with closed, proprietary models, and the open-weight models that we tested are too weak at this task to be usefully analyzed (see appendix section E). But such explanations are certainly an important direction for future work, which we are eager to explore. This being said, we do believe that our empirical results are an important first step in setting the stage for experiments that get at such explanations. In order to investigate the reasoning behind such shortcuts, it is necessary to first establish evaluation techniques that detect them meaningfully, which we hope to contribute with our work.
>
> $\textbf{How sensitive are the results to prompt design?}$
>
> This is an interesting question, however, we did not experiment with different prompts. For the sake of comparison, we used the same prompt as used by the ARC-Prize Foundation in their report about o3's "breakthrough" on abstract reasoning: https://arcprize.org/blog/oai-o3-pub-breakthrough. As we mention in our Limitations statement, it was not our focus here to investigate approaches to increase intended abstractions, but instead to offer a nuanced perspective on strong prior results. Investigating prompt sensitivity, as well as the effect of in-context demonstrations, is another topic for future research.
>
> $\textbf{Have you examined whether certain concept groups systematically elicit shortcut reasoning, suggesting task-type bias?}$
>
> We have spent some time computing the corresponding statistics and investigating the reasoning traces. We have added a per-concept analysis of shortcut reasoning to the revised manuscript (appendix J). We note that the concepts TopBottom3D and CleanUp have the highest share of correct-unintended rules, often relying on heuristics, such as shape density or bounding box overlaps. However, there are also general-purpose heuristics that are used for various concepts. This is indeed an interesting direction which we believe strengthened our contribution.

---

> > ### Comment · Reviewer_abxG · 2025-11-24
> > **Thanks for the response**
> >
> > The author response has addressed most of my concerns. I will retain my score recommending acceptance.

---

### Author Response · Authors · 2025-12-03
**Authors' summary of reviewer concerns and our responses**

To the AC: We believe that we have addressed the concerns of all five reviewers.  Their main concerns, and our responses, are as follows:

*Interpretability of rule faithfulness:* We performed an additional analysis of whether rules are  faithful by assessing alignment of rules and output grids for every task in three experimental settings.  We found that the generated rules correctly described the output grids in over 90% of the cases for each setting.  The results are given in Table 2 in our revised paper.

*Procedure for rule annotation:*  In our revised paper, we give a more detailed account of how rule evaluations were performed and verified by our team.

*Lack of mechanistic explanation for “shortcuts”:*  We responded that a mechanistic or representational investigation is not possible with closed, proprietary models, and the open-weight models that we tested are too weak at this task to be usefully analyzed.  However,  our empirical results and evaluation methodology are necessary steps for future mechanistic explanations.

*Systematic usage of “shortcuts” for specific concepts:*
We responded by adding a per-concept analysis of shortcut reasoning to the revised manuscript (appendix J). We noted that the concepts TopBottom3D and CleanUp have the highest share of correct-unintended rules, often relying on heuristics, such as shape density or bounding box overlaps. However, there are also general-purpose heuristics that are used for various concepts.

*No testing of prompt sensitivity:* We responded that, for the sake of comparison, we used the same zero-shot prompt as used by the ARC-Prize Foundation in their report about o3's "breakthrough" on abstract reasoning, as our focus was to investigate the positive results obtained with this particular prompt.  Investigating prompt sensitivity, as well as in-context examples, are topics for future work.

*Additional literature review:* In our revised paper, we have added a related work section with expanded discussion of relevant literature.

*Why ConceptARC?:* In our revised paper, we have made clearer the reason for using the ConceptARC dataset for this study—the fact that each task focuses on a specific human “core-knowledge” concept allows for more straightforward analysis of generated rules.

*Analyzing failure modes:* We responded that in our paper we analyze failure modes of both generated rules and output grids at a high level.  We were unable to include a detailed examination of all failure modes, as there are many ways in which LLMs can use superficial features in their solutions.  We did not analyze reasoning traces due to limited time and the fact that the provided traces are known to be incomplete. We agree that an in-depth analysis of failure modes is an important direction for future work.

*Limited novelty:* While it is known that reasoning models can use superficial features or shortcuts, this has not been shown before in the context of ARC-domain tasks.  We believe that this is an important result in the face of many claims of success of reasoning models on “abstract reasoning” using accuracy on ARC as the evaluation method.  The assumption that accuracy alone is sufficient to evaluate such models needs to be questioned.  In addition, ours is the only study we know of that simultaneously examines the effects of input modality, reasoning effort, and Python tool use on tasks in the ARC domain.

Three of five reviewers replied to our rebuttals, two stating that they retain their "accept" rating, and one (who recommended rejection) asking us to cite several papers; we responded with clarification as to how those works differ in scope from our contributions.   Because the reviewers are now prevented from responding and modifying their scores, we hope that the AC will consider whether we sufficiently addressed their concerns with our responses and paper revisions.

---

### Meta-Review · Area_Chair_Lziy · 2026-01-04

**Summary:**

The submission receives initial scores of 6, 0, 8, 2, and 2. The AC considers the motivation of studying abstract reasoning in o3-like reasoning models to be timely, interesting, and meaningful. However, the current study lacks sufficient rigor and completeness, and does not yield clear or well-supported conclusions. In particular, the analyses remain largely descriptive and fall short of providing convincing evidence or actionable insights that could inform the design of future reasoning benchmarks or models.

Based on the current version, the AC recommends rejection. The AC encourages the authors to substantially revise the paper in response to the reviewers’ comments and strengthen the depth and clarity of the analysis in a future submission.

**Reviewer Concerns:**

The reviewers raise several concerns. Three reviewers respond to the rebuttal, and the remaining key and shared issues are summarized below.

1. Reviewers CGz5, TPSZ, and RFre: The reviewers note that the novelty is limited, as the main insights are already well known (CGz5), or the conclusions are largely descriptive rather than providing systematic guidance for model or benchmark design (TPSZ and RFre).

    The rebuttal emphasizes the paper’s novelty in being among the first to study advanced reasoning models such as o3 on ARC-style tasks, going beyond final-answer accuracy and introducing intended abstraction to analyze shortcut behaviors.

    The AC agrees and finds that most reviewers also agree that the research motivation is timely and meaningful, and acknowledge the value of the current contributions. The AC also believes that a paper can be valuable without introducing a new benchmark or method, provided that the analyses and insights are sufficiently strong. However, the AC shares the reviewers’ concerns that the current version lacks depth in analysis and insight. The paper largely focuses on “what happens,” without sufficiently addressing “why it happens” or “how these findings should inform the design of future reasoning evaluations and models.” In addition, the observations remain largely qualitative and are not developed into new analytical frameworks.

    Beyond deeper mechanistic analysis, the AC also recommends that the authors substantially reorganize the paper with a clearer logical structure, explicitly connecting motivation, analysis methodology, observed phenomena, and resulting insights for each major conclusion.

2. Reviewer CGz5 and RFre: The reviewer notes missing literature review and comparisons to recent work. The rebuttal adds a corresponding section and provides detailed comparisons in the response. The AC recommends that the added material be incorporated into a future revision.
3. Reviewer RFre: The reviewer points out that performance in visual reasoning may be influenced by perceptual encoding factors in addition to abstraction and rule application, which are not fully considered in the paper. The rebuttal adds an analysis by dividing the data into two subsets. However, the AC considers that this issue warrants deeper discussion and analysis beyond a simple subset split.

**Reviewer Scores:**

Reviewers abxG, CGz5, and sHsM indicate in their responses that they will maintain their initial scores of 6, 0, and 8, respectively. Reviewers TPSZ and RFre acknowledge the motivation of the paper but maintain concerns about the strength of its contributions, as discussed in Concern 1. These concerns are not fully addressed in the rebuttal, and they are therefore likely to maintain their scores of 2 and 2.

---

### Decision · Program_Chairs · 2026-01-26

Reject